# Robust Contextual Linear Bandits

## Abstract

Model misspecification is a major consideration in applications of statistical methods and machine learning. However, it is often neglected in contextual bandits. This paper studies a common form of misspecification, an inter-arm heterogeneity that is not captured by context. To address this issue, we assume that the heterogeneity arises due to arm-specific random variables, which can be learned. We call this setting a *robust contextual bandit*. The arm-specific variables explain the unknown inter-arm heterogeneity, and we incorporate them in the robust contextual estimator of the mean reward and its uncertainty. We develop two efficient bandit algorithms for our setting: a UCB algorithm called `RoLinUCB` and a posterior-sampling algorithm called `RoLinTS`. We analyze both algorithms and bound their $n$-round Bayes regret. Our experiments show that `RoLinTS` is comparably statistically efficient to the classic methods when the misspecification is low, more robust when the misspecification is high, and significantly more computationally efficient than its naive implementation.

## 1 Introduction

A *stochastic contextual bandit* (Auer et al., 2002; Li et al., 2010; Lattimore & Szepesvari, 2019) is an online learning problem where a *learning agent* sequentially interacts with an environment over $n$ rounds. In each round, the agent observes *context*, pulls an *arm* conditioned on the context, and receives a corresponding *stochastic reward*. Contextual bandits have many applications in practice, such as in personalized recommendations (Li et al., 2010; Jeunen & Goethals, 2021). This is because the mean rewards of the arms are tied together through known context and learned model parameters. Thus the contextual approach can be more statistically efficient than a naive multi-armed bandit solution (Auer et al., 2002; Agrawal & Goyal, 2012). The linear model, where the mean reward of an arm is the dot product of its context and an unknown parameter, is versatile and popular (Dani et al., 2008; Rusmevichientong & Tsitsiklis, 2010; Abbasi-Yadkori et al., 2011; Agrawal & Goyal, 2013), and we consider it in this work.

There are two common approaches to using linear models in contextual bandits. One maintains a separate parameter per arm (Section 3.1 in Li et al. (2010)). While this approach can learn complex models, it is not very statistically efficient because the arm parameters are not shared. This is especially important when each arm is pulled a different number of times. The other approach maintains a single shared parameter for all arms. While this approach can be statistically efficient, it is more rigid and likely to fail due to model misspecification; when the optimal arm under the assumed model is not the actual optimal arm.

To address the above issues, we propose a new contextual linear model. This model assumes that the mean reward of an arm is a dot product of its context and an unknown shared parameter, which is offset by an arm-specific variable. This approach is statistically efficient because the model parameter is shared by all arms; yet flexible because the arm-specific variables can address model misspecification. We call this setting a *robust contextual linear bandit*. To provide an efficient solution to the problem, we assume that the arm-specific variables are random and drawn from a distribution known by the agent. This allows us to develop a joint estimator of the shared parameter and the arm-specific variables, which interpolates between the two and also uses the context.

One motivating example for our setting are recommender systems, where redeach item is considered as an arm and the features of an item cannot explain all information about the item, such as its intrinsic popularity (Koren et al., 2009). This is why the so-called behavioral features, the features that summarize the past engagement with the item, exist. The intrinsic popularity can be viewed as the average engagement, click or purchase rate, in the absence of

any other information. The item features then offset the engagement, either up or down, depending on their affinity. For instance, a feature representing the position of the item in the recommended list would have a negative weight, meaning that lower ranked items are less likely to be clicked, no matter how engaging they are. Luckily, our method can address it by defining item-specific variables to explain the intrinsic popularity.

We make the following contributions. First, we propose robust contextual linear bandits, where the model misspecification can be learned using arm-specific variables (Section 3). Under the assumption that the variables are random, both the Bayesian and random-effect viewpoints can be used to derive efficient joint estimators of the shared model parameter and arm-specific variables. We derive the estimators in Section 4, and show how to incorporate them in the estimate of the mean arm reward and its uncertainty. Second, we propose *upper confidence bound (UCB)* and *Thompson sampling (TS)* algorithms for this problem, RoLinUCB and RoLinTS (Section 5). Both algorithms are computationally efficient and robust to model misspecification. We analyze both algorithms and derive upper bounds on their $n$-round Bayes regret (Section 6). Our proofs rely on analyzing an equivalent linear bandit, and the resulting regret bounds improve in constants due to the special covariance structure of learned parameters. Our algorithms are also significantly more computationally efficient than naive implementations, which take $O((d + K)^3)$ time for $d$ dimensions and $K$ arms, instead of our $O(d^2(d + K))$. Finally, we evaluate RoLinTS on both synthetic and real-world problems. We observe that RoLinTS is comparably statistically efficient to the classic methods when the misspecification is low, more robust when the misspecification is high, and significantly more computationally efficient than its naive implementation.

## 2 Related Work

Our model is related to a hybrid linear model (Section 3.2 in Li et al. (2010)) with shared and arm-specific parameters. Unlike the hybrid linear model, where the coefficients of some features are shared by all arms while the others are not, we introduce arm-specific random variables to capture the model misspecification. We further study the impact of this structure on regret and propose an especially efficient implementation for this setting. Another related work is hLinUCB of Wang et al. (2016). hLinUCB is a variant of LinUCB that learns a portion of the feature vector and we compare to it in Section 7.

Due to our focus on robustness, our work is related to misspecified linear bandits. Ghosh et al. (2017) proposed an algorithm that switches from a linear to multi-armed bandit algorithm when the linear model is detected to be misspecified. Differently from this work, we adapt to misspecification. We do not compare to this algorithm in the main paper because it is non-contextual; and thus would have a linear regret in our setting. However, we conduct a comparison in a non-textual setting in Appendix C. Foster et al. (2020) and Krishnamurthy et al. (2021) proposed oracle-efficient algorithms that reduce contextual bandits to online regression, and are robust to misspecification. Since Safe-FALCON of Krishnamurthy et al. (2021) is an improvement upon Foster et al. (2020), we discuss it in more detail. Safe-FALCON is more general than our approach because it does not make any distributional assumptions. On the other hand, it is very conservative in our setting because of inverse gap weighting. We compare to it in Section 7. Takemura et al. (2021) proposed the first algorithm for the misspecified linear contextual bandit problem without knowledge of the approximation parameter. This algorithm is an extension of SupLinUCB (Chu et al., 2011) and, at round $t$ iteratively narrows down the set of arms in each stage until an arm is chosen from the set. It operates under the assumption that the approximation error is bounded. In contrast to their approach, our algorithm takes a Bayesian approach to estimate the approximation error in a Bayesian framework and subsequently constructs the corresponding UCB or TS algorithm. Dong & Yang (2023) investigated the role of sparsity in improving misspecified bandit learning, and Zhang et al. (2023) studied how the interplay between the misspecification level and the sub-optimality gap affects the performance of linear contextual bandits. These theoretical studies differs from ours in terms of motivation and content. In our work, we focus on modeling the misspecified component and developing efficient algorithms tailored to address this aspect. Finally, Bogunovic et al. (2021) and Ding et al. (2022) proposed linear bandit algorithms that are robust to adversarial noise attack. The notion of robustness in these works is very different from ours.

Our work is also related to random-effect bandits (Zhu & Kveton, 2022). As in Zhu & Kveton (2022), we assume that each arm is associated with a random variable that can help with explaining its unknown mean reward. Zhu & Kveton (2022) used this structure to design a bandit algorithm that is comparably efficient to TS without knowing the prior. Their algorithm is UCB not contextual. A similar idea was explored by Wan et al. (2022) and applied to structured bandits. This work is also non-contextual. Wan et al. (2021) assumed a hierarchical structure over tasks and modeled

inter-task heterogeneity. We focus on a single task and model inter-arm heterogeneity. Our work is also related to recent papers on hierarchical Bayesian bandits (Kveton et al., 2021; Basu et al., 2021; Hong et al., 2022). All of these papers considered a similar graphical model to Wan et al. (2021) and therefore model inter-task heterogeneity.

## 3 Robust Contextual Linear Bandits

We adopt the following notation. For any positive integer $n$, we denote by $[n]$ the set $\{1, \ldots, n\}$. We let $\mathbb{1}\{\cdot\}$ be the indicator function. For any matrix $\mathbf{M} \in \mathbb{R}^{d \times d}$, the maximum eigenvalue is $\lambda_1(\mathbf{M})$ and the minimum is $\lambda_d(\mathbf{M})$. The big O notation up to logarithmic factors is $\tilde{O}$.

We consider a *contextual linear bandit* (Li et al., 2010), where the relationship between the mean reward of an arm and its context is represented by a model. In round $t \in [n]$, an agent pulls one of $K$ arms with feature vectors $\mathbf{x}_{i,t} \in \mathbb{R}^d$ for $i \in [K]$. The vector $\mathbf{x}_{i,t}$ summarizes information specific to arm $i$ in round $t$ and we call it a *context*. Compared to context-free bandits (Lai & Robbins, 1985; Auer et al., 2002; Agrawal & Goyal, 2012), contextual linear bandits have more practical applications because they model the reward as a function of context. For instance, in online advertising, the arms would be different ads, the context would be user features, and the contextual bandit agent would pull arms according to user features (Li et al., 2010; Agrawal & Goyal, 2013). More formally, in round $t$, the agent pulls arm $I_t \in [K]$ based on context and rewards in past rounds; and receives the reward of arm $I_t$, $r_{I_t,t}$, whose mean reward depends on the context $\mathbf{x}_{I_t,t}$. Since the number of contexts is large, the agent assumes some generalization model, such as that the mean reward is linear in $\mathbf{x}_{i,t}$ and some unknown parameter. When this model is incorrectly specified, the contextual bandit algorithm may perform poorly.

To improve the robustness of contextual linear bandits to misspecification, we introduce a Bayesian hierarchical modeling assumption. Specifically, the reward $r_{i,t}$ of arm $i$ in round $t$ is generated as

$$r_{i,t} = \mu_{i,t} + \epsilon_{i,t}, \tag{1}$$

$$\mu_{i,t} = \mathbf{x}_{i,t}^\top \boldsymbol{\theta} + v_i, \tag{2}$$

$$\boldsymbol{\theta} \sim P_\theta(\mathbf{0}, \lambda^{-1}\mathbf{I}_d), \tag{3}$$

$$v_i \sim P_v(0, \sigma_0^2), \tag{4}$$

$$\epsilon_{i,t} \sim P_\epsilon(0, \sigma^2). \tag{5}$$

Here $\mu_{i,t}$ and $\epsilon_{i,t}$ are the mean reward and reward noise, respectively, of arm $i$ in round $t$. The mean reward $\mu_{i,t}$ has two terms: a linear function of context $\mathbf{x}_{i,t}$ and parameter $\boldsymbol{\theta} \in \mathbb{R}^d$ shared by all arms, and the inter-arm heterogeneity $v_i \in \mathbb{R}$, which is an unobserved arm-specific random variable. The distributions of $\boldsymbol{\theta}$, $v_i$, and $\epsilon_{i,t}$ are denoted by $P_\theta$, $P_v$, and $P_\epsilon$; and their hyper-parameters are $\lambda$, $\sigma_0^2$, and $\sigma^2$. We call our model a *robust contextual linear bandit* because $v_i$ makes it robust to the misspecification due to context. Our model can be viewed as an instance of unobserved-effect models commonly used in panel and longitudinal data analyses (Wooldridge, 2001; Diggle et al., 2002). For brevity, and since we only study linear models, we often call our model a *robust contextual bandit*.

Our goal is to design an algorithm that minimizes its regret with respect to the optimal arm-selection strategy. The *$n$-round Bayes regret $R(n)$* of an agent is defined as

$$R(n) = \mathbb{E}\left[\sum_{t=1}^n \mu_{I_t^*,t} - \mu_{I_t,t}\right], \tag{6}$$

where $I_t^*$ is the arm with highest mean reward in round $t$ and $I_t$ is the pulled arm in round $t$. The expectation is under the randomness of $I_t$ and $I_t^*$; and those of $\boldsymbol{\theta}$, $v_i$, and $\epsilon_{i,t}$. Given that our model is within a Bayesian setting, this paper focuses on the Bayesian regret, wherein the expectation is taken with respect to the distribution over the inter-arm heterogeneity. Through an analysis of Bayesian regret, we quantify the expected regret concerning $v_i$.

### 3.1 Discussion

We introduce an unobserved effect $v_i$, which can be interpreted as capturing the characteristics of arm $i$ that is not explained by context, but is assumed not to change over $n$ rounds. We call it the *inter-arm heterogeneity*. For example,

in online advertising, the arms would be different ads and the context would be user features. In this problem, $v_i$ may contain unobserved ad characteristics, such as its intrinsic quality, that can be viewed as roughly constant.

We assume that the parameter $\boldsymbol{\theta}$ is shared by all arms, while the inter-arm heterogeneity is modeled by arm-specific variables. From the statistical-efficiency viewpoint, this model reduces the number of parameters compared to modeling arms separately, and therefore increases statistical efficiency. Li et al. (2010) proposed hybrid linear models, where the coefficients of some features are shared by all arms while the others are arm-specific. However, choosing features to share may be challenging in practice. From the practical viewpoint, it is more convenient to apply the robust contextual bandit, as it avoids the challenging choice of the shared features. In particular, the model is still flexible enough because it uses the unobserved effect $v_i$ to capture inter-arm heterogeneity, information not explained by the context. For instance, imagine a contextual recommendation problem with $K$ arms, where arms represent items. In addition to what the item and user features can explain, there may still be item-specific biases (Koren et al., 2009). The main difference from the work of Li et al. (2010) is that our approach is a two-level Bayesian model. We model the linear relationship between rewards and contexts. At the arm level, we incorporate a Bayesian assumption, positing that the inter-arm heterogeneity $v_i$ follows the distribution in (4). This assumption translates to how we estimate $\mu_{i,t}$, which is a weighted average of the linear model and the inter-arm heterogeneity. These weights are adjusted to reflect the relative magnitudes of the lack of fit of the linear model and the variance of the inter-arm heterogeneity. We present more details in Section 4.

## 4 Estimation

This section introduces our estimators for robust contextual bandits. In Section 4.1, we derive the estimators of $\boldsymbol{\theta}$ and $v_i$ for $i \in [K]$. In Section 4.2, we derive the estimator of $\mu_{i,t} = \mathbf{x}_{i,t}^\top \boldsymbol{\theta} + v_i$ and its uncertainty.

### 4.1 Maximum a Posteriori Estimation of $\theta$ and $v_i$

Fix round $t$. Let $\mathcal{T}_{i,t}$ be the set of rounds where arm $i$ is pulled by the beginning of round $t$ and $n_{i,t} = |\mathcal{T}_{i,t}|$ be the size of $\mathcal{T}_{i,t}$. Let $\mathbf{r}_{i,t} = (r_{i,\ell})_{\ell \in \mathcal{T}_{i,t}}^\top$ be the column vector of rewards obtained by pulling arm $i$, $\boldsymbol{\epsilon}_{i,t} = (\epsilon_{i,\ell})_{\ell \in \mathcal{T}_{i,t}}^\top$ be the column vector of the corresponding reward noise, and $\mathbf{X}_{i,t} = (\mathbf{x}_{i,\ell})_{\ell \in \mathcal{T}_{i,t}}^\top$ be a $n_{i,t} \times d$ matrix with the corresponding contexts. From (1) and (2),

$$\mathbf{r}_{i,t} = \mathbf{X}_{i,t}\boldsymbol{\theta} + v_i \mathbf{1}_{n_{i,t}} + \boldsymbol{\epsilon}_{i,t},$$

where $\mathbf{1}_k$ is a vector of length $k$ whose all entries are one. The covariance matrix $\mathbf{V}_{i,t}$ for the vector $\mathbf{r}_{i,t}$ is given by $\mathbf{V}_{i,t} = \sigma_0^2 \mathbf{1}_{n_{i,t}} \mathbf{1}_{n_{i,t}}^\top + \sigma^2 \mathbf{I}_{n_{i,t}}$, where $\mathbf{I}_k$ is the identify matrix of size $k \times k$. The terms $\sigma_0^2 \mathbf{1}_{n_{i,t}} \mathbf{1}_{n_{i,t}}^\top$ and $\sigma^2 \mathbf{I}_{n_{i,t}}$ represent the randomness from $v_i$ and $\boldsymbol{\epsilon}_{i,t}$, respectively. By the Woodbury matrix identity,

$$\mathbf{V}_{i,t}^{-1} = \sigma^{-2}\mathbf{I}_{n_{i,t}} - \sigma^{-2} n_{i,t}^{-1} w_{i,t} \mathbf{1}_{n_{i,t}} \mathbf{1}_{n_{i,t}}^T. \tag{7}$$

Assuming that $P_\theta$, $P_v$, $P_\epsilon$ are Gaussian, the *maximum a posteriori (MAP)* estimation is equivalent to minimizing the following loss function

$$L(v_1, \cdots, v_K, \boldsymbol{\theta}) = \sum_{i=1}^{K} \left[ \sigma^{-2}\|\mathbf{r}_{i,t} - \mathbf{X}_{i,t}\boldsymbol{\theta} - v_i\|^2 + \sigma_0^{-2}v_i^2 \right] + \lambda\|\boldsymbol{\theta}\|^2 \tag{8}$$

with respect to $(v_i)_{i\in[K]}$ and $\boldsymbol{\theta}$, where $\|\cdot\|$ is the Euclidean norm. The term $\sigma^{-2}\sum_{i=1}^{K}\|\mathbf{r}_{i,t} - \mathbf{X}_{i,t}\boldsymbol{\theta} - v_i\|^2$ is from the conditional likelihood of $\mathbf{r}_{i,t}$ given $(v_i)_{i\in[K]}$ and $\boldsymbol{\theta}$. The regularization term $\sigma_0^{-2}\sum_{i=1}^{K}v_i^2$ is from the prior of $(v_i)_{i\in[K]}$ in (4). The other term $\lambda\|\boldsymbol{\theta}\|^2$ is from the prior of $\boldsymbol{\theta}$ in (3).

Differentiating $L(v_1, \cdots, v_K, \boldsymbol{\theta})$ with respect to $v_i$ and putting it equal to zero, $v_i$ is estimated by

$$\tilde{v}_{i,t} = w_{i,t}(\bar{r}_{i,t} - \bar{\mathbf{x}}_{i,t}^\top\boldsymbol{\theta}), \tag{9}$$

where $\bar{r}_{i,t} = n_{i,t}^{-1}\sum_{\ell\in\mathcal{T}_{i,t}} r_{i,\ell}$ is the average reward of arm $i$ up to round $t$, $\bar{\mathbf{x}}_{i,t} = n_{i,t}^{-1}\sum_{\ell\in\mathcal{T}_{i,t}} \mathbf{x}_{i,\ell}$ is the average context associated with the pulls of arm $i$ up to round $t$, and

$$w_{i,t} = \frac{\sigma_0^2}{\sigma_0^2 + \sigma^2/n_{i,t}} \tag{10}$$

is a weight that interpolates between the context and the arm-specific variable. We discuss its role in Section 4.2.

Let $\mathbf{u}_{i,t} = \mathbf{r}_{i,t} - \mathbf{X}_{i,t}\boldsymbol{\theta}$ and $\bar{u}_{i,t} = n_{i,t}^{-1} \sum_{\ell \in \mathcal{T}_{i,t}} u_{i,\ell}$, where $u_{i,\ell}$ is the $\ell$-th element of $\mathbf{u}_{i,t}$. Inserting (9) into (8), it follows that

$$
\begin{aligned}
L(\boldsymbol{\theta}) &= \sum_{i=1}^{K} \left[ \sigma^{-2}\|\mathbf{u}_{i,t} - w_{i,t}\bar{u}_{i,t}\|^2 + \sigma_0^{-2}w_{i,t}^2\bar{u}_{i,t}^2 \right] + \lambda\|\boldsymbol{\theta}\|^2 \\
&= \sigma^{-2} \sum_{i=1}^{K} \left[ \|\mathbf{u}_{i,t}\|^2 - n_{i,t}w_{i,t}\bar{u}_{i,t}^2 \right] + \lambda\|\boldsymbol{\theta}\|^2 \\
&= \sum_{i=1}^{K} \left[ (\mathbf{r}_{i,t} - \mathbf{X}_{i,t}\boldsymbol{\theta})^\top \mathbf{V}_{i,t}^{-1}(\mathbf{r}_{i,t} - \mathbf{X}_{i,t}\boldsymbol{\theta}) \right] + \lambda\|\boldsymbol{\theta}\|^2 \,,
\end{aligned}
$$

where the last step is from (7).

To obtain the MAP estimate of $\boldsymbol{\theta}$, we minimize $L(\boldsymbol{\theta})$ with respect to $\boldsymbol{\theta}$ and get

$$
\hat{\boldsymbol{\theta}}_t = \left( \lambda\mathbf{I}_d + \sum_{i=1}^{K} \mathbf{X}_{i,t}^\top \mathbf{V}_{i,t}^{-1}\mathbf{X}_{i,t} \right)^{-1} \sum_{i=1}^{K} \mathbf{X}_{i,t}^\top \mathbf{V}_{i,t}^{-1}\mathbf{r}_{i,t} \,. \tag{11}
$$

To obtain the MAP estimate of $v_i$, we insert $\hat{\boldsymbol{\theta}}_t$ into (9),

$$
\hat{v}_{i,t} = w_{i,t}(\bar{r}_{i,t} - \bar{\mathbf{x}}_{i,t}^\top\hat{\boldsymbol{\theta}}_t) \,. \tag{12}
$$

## 4.2 Prediction of $\mu_{i,t}$ and Its Uncertainty

Based on (11) and (12), the estimated mean reward of arm $i$ in context $\mathbf{x}_{i,t}$ in round $t$ is

$$
\hat{\mu}_{i,t} = \mathbf{x}_{i,t}\hat{\boldsymbol{\theta}}_t + w_{i,t}(\bar{r}_{i,t} - \bar{\mathbf{x}}_{i,t}^\top\hat{\boldsymbol{\theta}}_t) \,. \tag{13}
$$

In (2), $v_i$ represents the inter-arm heterogeneity. It is the arm-specific effect that cannot be explained by context. This effect is estimated in (13) using $w_{i,t}(\bar{r}_{i,t} - \bar{\mathbf{x}}_{i,t}^\top\hat{\boldsymbol{\theta}}_t)$. Now consider $w_{i,t}$ in (10). If the arm has not been pulled, $n_{i,t} = 0$ and $w_{i,t} = 0$. Therefore, $\hat{\mu}_{i,t} = \mathbf{x}_{i,t}^\top\hat{\boldsymbol{\theta}}_t$. Similarly, if the arm has not been pulled often, $\hat{\mu}_{i,t}$ is close to $\mathbf{x}_{i,t}\hat{\boldsymbol{\theta}}_t$. This means that the prediction $\hat{\mu}_{i,t}$ is statistically efficient for small $n_{i,t}$. This is helpful in the initial rounds when there are only a few observations of arms.

We further explain (13) by rewriting it as

$$
\hat{\mu}_{i,t} = w_{i,t}\bar{r}_{i,t} + (\mathbf{x}_{i,t} - w_{i,t}\bar{\mathbf{x}}_{i,t})^\top \hat{\boldsymbol{\theta}}_t \,. \tag{14}
$$

Here $\hat{\mu}_{i,t}$ is a weighted estimator of two terms: the sample mean of arm $i$, $\bar{r}_{i,t}$, and additional calibration from contexts $(\mathbf{x}_{i,t} - w_{i,t}\bar{\mathbf{x}}_{i,t})^\top\hat{\boldsymbol{\theta}}_t$. The weight is $w_{i,t}$. When $n_{i,t} \to \infty$, $w_{i,t} \to 1$ and $\hat{\mu}_{i,t} \to \bar{r}_{i,t} + (\mathbf{x}_{i,t} - \bar{\mathbf{x}}_{i,t})^\top\hat{\boldsymbol{\theta}}_t$. This shows why our prediction $\hat{\mu}_{i,t}$ is robust. Informally, it uses $\bar{r}_{i,t}$ as a baseline and corrects it using context as $(\mathbf{x}_{i,t} - \bar{\mathbf{x}}_{i,t})^\top\hat{\boldsymbol{\theta}}_t$. Therefore, we reduce the reliance on the contextual model by automatically balancing the contextual and multi-armed bandits. This is why we call our framework a robust contextual bandit.

The prediction $\hat{\mu}_{i,t}$ can also degenerate to that of a contextual linear bandit (Rusmevichientong & Tsitsiklis, 2010; Agrawal & Goyal, 2013). More specifically, $w_{i,t} \to 0$ as $\sigma_0^2 \to 0$, and then $\hat{\mu}_{i,t}$ approaches

$$
\hat{\mu}_{i,t}^{\text{lin}} = \mathbf{x}_{i,t} \left( \lambda\mathbf{I}_d + \sum_{i=1}^{K} \mathbf{X}_{i,t}^\top\mathbf{X}_{i,t} \right)^{-1} \sum_{i=1}^{K} \mathbf{X}_{i,t}^\top\mathbf{r}_{i,t} \,,
$$

which is the prediction of a simple linear model without inter-arm heterogeneity. This observation is important because it shows that our framework is as general as the contextual linear bandit.

Now we characterize the uncertainty of $\hat{\mu}_{i,t}$ in (14) for efficient exploration. We measure the uncertainty by the mean squared error $\mathrm{E}[(\hat{\mu}_{i,t} - \mu_{i,t})^2]$. Let $\bar{\epsilon}_{i,t} = n_{i,t}^{-1} \sum_{\ell \in \mathcal{T}_{i,t}} \epsilon_{i,\ell}$ and $\mathbf{M}_t = \lambda \mathbf{I}_d + \sum_{i=1}^{K} \mathbf{X}_{i,t}^{\top} \mathbf{V}_{i,t}^{-1} \mathbf{X}_{i,t}$. A direct calculation shows that

$$
\begin{aligned}
&\mathrm{E}[(\hat{\mu}_{i,t} - \mu_{i,t})^2] \\
=&\mathrm{E}[((w_{i,t} - 1)v_i + w_{i,t}\bar{\epsilon}_{i,t} + (\mathbf{x}_{i,t} - w_{i,t}\bar{\mathbf{x}}_{i,t})^{\top}(\hat{\boldsymbol{\theta}}_t - \boldsymbol{\theta}))^2] \\
=&\sigma_0^2(1 - w_{i,t}) + (\mathbf{x}_{i,t} - w_{i,t}\bar{\mathbf{x}}_{i,t})^{\top}\mathbf{M}_t^{-1}(\mathbf{x}_{i,t} - w_{i,t}\bar{\mathbf{x}}_{i,t}) \\
=:&\tau_{i,t}^2 \,,
\end{aligned}
\tag{15}
$$

where the last step is from $\mathrm{E}[(w_{i,t}v_i - v_i + w_{i,t}\bar{\epsilon}_{i,t})(\hat{\boldsymbol{\theta}}_t - \boldsymbol{\theta})^{\top}(\mathbf{x}_{i,t} - w_{i,t}\bar{\mathbf{x}}_{i,t})] = 0$ shown in Kachar & Harville (1984).

### 4.3 Computational Efficiency

We also investigate if the robust contextual bandit can be implemented as computationally efficiently as a contextual linear bandit. As discussed in Section 6.2, our model is equivalent to a linear model augmented by $K$ features, indicating which unobserved $v_i$ corresponds to arm $i$. The computational cost of posterior sampling or computing upper confidence bounds in this model is $O((d + K)^3)$ per round, due to inverting $(d + K) \times (d + K)$ precision matrices. On the other hand, the robust contextual bandit can be implemented as computationally efficiently as a contextual linear bandit with $d$ features, with $O(d^2(d + K))$ computational cost per round. Specifically, using (7),

$$
\begin{aligned}
\mathbf{X}_{i,t}^{\top}\mathbf{V}_{i,t}^{-1}\mathbf{X}_{i,t} &= \sigma^{-2}(\mathbf{X}_{i,t}^{\top}\mathbf{X}_{i,t} - w_{i,t}n_{i,t}\bar{\mathbf{x}}_{i,t}\bar{\mathbf{x}}_{i,t}^{\top}) \,, \\
\mathbf{X}_{i,t}^{\top}\mathbf{V}_{i,t}^{-1}\mathbf{r}_{i,t} &= \sigma^{-2}(\mathbf{X}_{i,t}^{\top}\mathbf{r}_{i,t} - w_{i,t}n_{i,t}\bar{\mathbf{x}}_{i,t}\bar{r}_{i,t}) \,.
\end{aligned}
$$

These identities can be used to rederive all statistics as

$$
\mathbf{M}_t = \lambda \mathbf{I}_d + \sigma^{-2} \sum_{i=1}^{K}(\mathbf{X}_{i,t}^{\top}\mathbf{X}_{i,t} - w_{i,t}n_{i,t}\bar{\mathbf{x}}_{i,t}\bar{\mathbf{x}}_{i,t}^{\top}) \,,
$$

$$
\hat{\boldsymbol{\theta}}_t = \mathbf{M}_t^{-1}\left[\sigma^{-2}\sum_{i=1}^{K}(\mathbf{X}_{i,t}^{\top}\mathbf{r}_{i,t} - w_{i,t}n_{i,t}\bar{\mathbf{x}}_{i,t}\bar{r}_{i,t})\right] \,,
\tag{16}
$$

$$
\hat{\mu}_{i,t} = w_{i,t}\bar{r}_{i,t} + (\mathbf{x}_{i,t} - w_{i,t}\bar{\mathbf{x}}_{i,t})^{\top}\hat{\boldsymbol{\theta}}_t \,,
\tag{17}
$$

$$
\tau_{i,t}^2 = \sigma_0^2(1 - w_{i,t}) + (\mathbf{x}_{i,t} - w_{i,t}\bar{\mathbf{x}}_{i,t})^{\top}\mathbf{M}_t^{-1}(\mathbf{x}_{i,t} - w_{i,t}\bar{\mathbf{x}}_{i,t}) \,.
\tag{18}
$$

The main cost in the above formulas is due to calculating $\mathbf{M}_t^{-1}$, which is $O(d^3)$ per round. All remaining operations are $O(d^2 K)$. Therefore, the computational cost of prediction in the robust contextual bandit is $O(d^2(d + K))$ and comparable to the contextual linear bandit.

## 5 Algorithms

*Upper confidence bounds (UCBs)* (Auer et al., 2002) and *Thompson sampling (TS)* (Thompson, 1933) are two popular bandit algorithm designs. We propose UCB and TS algorithms for robust contextual bandits based on the estimate of $\mu_{i,t}$ and its uncertainty (Section 4). The UCB algorithm is called RoLinUCB because it can be viewed as a robust variant of LinUCB (Abbasi-Yadkori et al., 2011). From Section 4, $\hat{\mu}_{i,t}$ and $\tau_{i,t}^2$ are the posterior mean and variance, respectively, of $\mu_{i,t}$ in round $t$. This observation motivates a posterior-sampling algorithm that uses the posterior of $\mu_{i,t}$. We call it RoLinTS because it can be viewed as a robust variant of LinTS (Agrawal & Goyal, 2013).

Both algorithms work as follows. Let the history at the beginning of round $t$ be all actions and observations of the agent up to that round, $H_t = (\mathbf{x}_{I_\ell,\ell}, I_\ell, r_{I_\ell,\ell})_{\ell=1}^{t-1}$. In round $t$, the algorithms observe context $\mathbf{x}_{i,t}$ of each arm $i$ and then compute the MAP estimate $\hat{\mu}_{i,t}$ of $\mu_{i,t}$ and its uncertainty $\tau_{i,t}^2$ conditioned on $H_t$. RoLinUCB pulls the arm with the highest upper confidence bound, $I_t = \arg\max_{i \in [K]} U_{i,t}$, where $U_{i,t} = \hat{\mu}_{i,t} + \sqrt{2\tau_{i,t}^2 \log n}$. RoLinTS samples $U_{i,t} \sim$

$\mathcal{N}(\hat{\mu}_{i,t}, \tau_{i,t}^2)$ and then pulls the arm with the highest mean reward under the posterior sample, $I_t = \arg\max_{i \in [K]} U_{i,t}$. After pulling arm $I_t$ and observing the corresponding reward, the algorithms update all statistics in Section 4.3. The pseudo-code of RoLinTS and RoLinUCB is presented in Algorithm 1.

---

**Algorithm 1** RoLinUCB and RoLinTS for robust contextual bandits.

---

1: **for** $t = 1, \ldots, n$ **do**
2:     **for** $i = 1, \ldots, K$ **do**
3:         Observe contexts $\mathbf{x}_{i,t}$
4:         Obtain $\hat{\mu}_{i,t}$ from (17) and $\tau_{i,t}^2$ from (18)
5:         Define

$$\texttt{RoLinUCB: } U_{i,t} = \hat{\mu}_{i,t} + \sqrt{2\tau_{i,t}^2 \log n}$$

$$\texttt{RoLinTS: } U_{i,t} \sim \mathcal{N}(\hat{\mu}_{i,t}, \tau_{i,t}^2)$$

6:     **end for**
7:     $I_t \leftarrow \arg\max_{i \in [K]} U_{i,t}$
8:     Pull arm $I_t$ and observe reward $r_{I_t,t}$
9:     $n_{I_t,t} \leftarrow n_{I_t,t} + 1$
10:    Update all statistics in Section 4.3
11: **end for**

---

## 6 Regret Analysis

We prove upper bounds on the $n$-round regret of RoLinUCB and RoLinTS. Similarly to random-effect bandits (Zhu & Kveton, 2022), $\hat{\mu}_{i,t}$ is the MAP estimate of $\mu_{i,t}$ given history $H_t$, under the assumptions that $P_\theta$, $P_v$, and $P_\epsilon$ are Gaussian distributions. Thus we adopt the Bayes regret (Russo & Van Roy, 2014) to analyze RoLinUCB and RoLinTS. Let the optimal arm in round $t$ be $I_t^* = \arg\max_{i \in [K]} \mu_{i,t}$. The regret is the difference between the rewards that we would have obtained by pulling the optimal arm $I_t^*$ and the rewards that we did obtain by pulling $I_t$ over $n$ rounds. The regret is formally defined in (6) and we bound it below.

**Theorem 1.** *Consider the robust contextual bandit where*

$$P_\theta = \mathcal{N}(\mathbf{0}, \lambda^{-1}\mathbf{I}_d), \ P_v = \mathcal{N}(0, \sigma_0^2), \ P_\epsilon = \mathcal{N}(0, \sigma^2).$$

*Let the hyper-parameters $\lambda$, $\sigma_0^2$, $\sigma^2$ be known by the learning agent. Let $\|\mathbf{x}_{i,t}\| \leq L$. Then the $n$-round Bayes regret of* RoLinUCB *and* RoLinTS *is bounded as*

$$R(n) \leq \sigma_{\max}\sqrt{2c(d+K)n\log(n)} + \sqrt{2/\pi}\sigma_{\max}K,$$

*where*

$$c = \log\left(1 + \frac{\sigma_{\max}^2 n}{\sigma^2(d+K)}\right) \bigg/ \log(1 + \sigma^{-2}\sigma_{\max}^2)$$

*and $\sigma_{\max}^2 = \sigma_0^2 + L^2\lambda^{-1}$.*

### 6.1 Discussion

Theorem 1 shows that the Bayes regret of both RoLinUCB and RoLinTS is $O(\sqrt{(\sigma_0^2 + L^2\lambda^{-1})(d+K)n})$ up to logarithmic factors. The dependence on horizon $n$ is optimal. The dependence on $d + K$ arises due to learning $d + K$ parameters in the equivalent linear bandit: $d$ for the linear model and one parameter per arm. This would be optimal in a general linear bandit. However, the dependence on $K$ does not vanish as the inter-arm heterogeneity diminishes. This limitation is inherent in our analysis, as we employ the linear bandit framework with $d + K$ parameters. Empirically, we have observed that the dependence on $K$ would vanish as $\sigma_0^2 \rightarrow 0$ (Appendix C). The structure of our

problem is captured by constant $\sigma_0^2 + L^2\lambda^{-1}$. The regret increases when the shared parameter $\boldsymbol{\theta}$ is more uncertain, $\lambda$ is low; when the inter-arm heterogeneity is high, $\sigma_0$ is high; and when the feature vectors of arms are long, $L$ is high.

Theorem 1 is proved under the assumption that the hyper-parameters $\lambda$, $\sigma_0^2$, $\sigma^2$ are known to the learning agent. We evaluated `RoLinTS` empirically to hyper-parameter misspecification (Figure 3 in Appendix C). Notably, our findings demonstrate that the algorithm maintains its performance.

Our analysis in Section 6.2 improves upon a trivial linear bandit analysis by using the structure of the posterior variance in (15). A trivial analysis, which would only use the structure in the prior covariance,

$$\|\mathbf{z}_{i,t}\|_{\boldsymbol{\Sigma}_t}^2 \leq \|\mathbf{z}_{i,t}\|_{\boldsymbol{\Sigma}_0}^2 \leq \max\left\{\sigma_0^2, \lambda^{-1}\right\}(L^2+1),$$

would replace the factor $\sigma_0^2 + L^2\lambda^{-1}$ in our regret bound with $\max\left\{\sigma_0^2, \lambda^{-1}\right\}(L^2+1)$. Note that

$$\sigma_0^2 + L^2\lambda^{-1} \leq \max\left\{\sigma_0^2, \lambda^{-1}\right\}(L^2+1)$$

for any $\lambda$, $\sigma_0$, and $L$; and hence our analysis is always an improvement. This improvement can be significant when the parameter $\boldsymbol{\theta}$ is nearly certain and $K \gg d$. In this case, our bound approaches that of a $K$-armed Bayesian bandit with prior $\mathcal{N}(0, \sigma_0^2)$, which is $O(\sqrt{\sigma_0^2 Kn})$; and the other bound is $O(\sqrt{\sigma_0^2 L^2 Kn})$, where $L^2$ could be $O(d)$.

Beyond improvements in regret, the structure of our problem can be used to get major improvements in computational efficiency (Section 4.3), from $O((d+K)^3)$ time per round for a naive implementation to $O(d^2(d+K))$.

## 6.2 Proof of Theorem 1

First, we note that (2) can be rewritten as a single linear model by augmenting features. Specifically, let $u \oplus v$ be the concatenation of vectors $u$ and $v$; and $e_i \in \mathbb{R}^K$ be an indicator vector of the $i$-th dimension, $e_{i,j} = \mathbb{1}\{i = j\}$. Using this notation, let $\mathbf{z}_{i,t} = \mathbf{x}_{i,t} \oplus e_i$ be the augmented feature vector of arm $i$ in round $t$ and $\boldsymbol{\gamma} = \boldsymbol{\theta} \oplus (v_i)_{i\in[K]}$ be the augment parameter vector. Then the model in (2), (3), and (4) can be expressed as a simple Bayesian linear regression model,

$$\mu_{i,t} = \mathbf{z}_{i,t}^\top \boldsymbol{\gamma}, \quad \boldsymbol{\gamma} \sim \mathcal{N}(\mathbf{0}, \boldsymbol{\Sigma}_0), \tag{19}$$

where $\boldsymbol{\Sigma}_0$ is a block-diagonal matrix. Its upper $d \times d$ block is $\lambda^{-1}\mathbf{I}_d$ and the lower $K \times K$ block is $\sigma_0^2\mathbf{I}_K$.

Let $\mathbf{r}_t = (r_{I_\ell,\ell})_{\ell\in[t-1]}$ be a column vector of all rewards up to round $t$ and $\mathbf{Z}_t = (\mathbf{z}_{I_\ell,\ell})_{\ell\in[t-1]}$ be a $(t-1) \times (d+K)$ matrix of augmented features up to round $t$. From (19), we have that

$$\boldsymbol{\gamma} \mid H_t \sim \mathcal{N}(\boldsymbol{\gamma}_t, \boldsymbol{\Sigma}_t),$$

where $\boldsymbol{\gamma}_t = \sigma^{-2}\boldsymbol{\Sigma}_t\mathbf{Z}_t^\top\mathbf{r}_t$ and $\boldsymbol{\Sigma}_t^{-1} = \boldsymbol{\Sigma}_0^{-1} + \sigma^{-2}\mathbf{Z}_t^\top\mathbf{Z}_t$. It follows that

$$\mu_{i,t} \mid H_t \sim \mathcal{N}(\mathbf{z}_{i,t}^\top\boldsymbol{\gamma}_t, \mathbf{z}_{i,t}^\top\boldsymbol{\Sigma}_t\mathbf{z}_{i,t}).$$

We prove in Lemma 1 that $\mu_{i,t} \mid H_t \sim \mathcal{N}(\hat{\mu}_{i,t}, \tau_{i,t}^2)$. Thus we can equivalently analyze the reformulation in (19).

Now we apply the general regret bound in Theorem 2 (Appendix B) to (19) and get

$$R(n) \leq \sigma_{\max}\sqrt{2c(d+K)n\log(1/\delta)} + \sqrt{2/\pi}\sigma_{\max}Kn\delta,$$

where

$$c = \log\left(1 + \frac{\sigma_{\max}^2 n}{\sigma^2(d+K)}\right) \Big/ \log(1 + \sigma^{-2}\sigma_{\max}^2)$$

and $\sigma_{\max} = \max_{i\in[K],\,t\in[n]}\|\mathbf{z}_{i,t}\|_{\boldsymbol{\Sigma}_t}$ is a problem-specific quantity that we bound next. Specifically, since $\|\mathbf{z}_{i,t}\|_{\boldsymbol{\Sigma}_t}^2 = \tau_{i,t}^2$, we have

$$\|\mathbf{z}_{i,t}\|_{\boldsymbol{\Sigma}_t}^2 \leq \sigma_0^2 + L^2\lambda_1(\mathbf{M}_t^{-1}) = \sigma_0^2 + L^2\lambda_d^{-1}(\mathbf{M}_t) \leq \sigma_0^2 + L^2\lambda^{-1}.$$

The first inequality is from the definition of $\tau_{i,t}^2$ in (15) and that $\|\mathbf{x}_{i,t}\| \leq L$. The second inequality is from the observation that both components of $\mathbf{M}_t$ in (15) are positive semi-definite (PSD). Specifically, all eigenvalues of $\lambda\mathbf{I}_d$ are $\lambda$ and each $\mathbf{X}_{i,t}^\top\mathbf{V}_{i,t}^{-1}\mathbf{X}_{i,t}$ is PSD because $\mathbf{V}_{i,t}$ is. To complete the proof, we set $\delta = 1/n$.

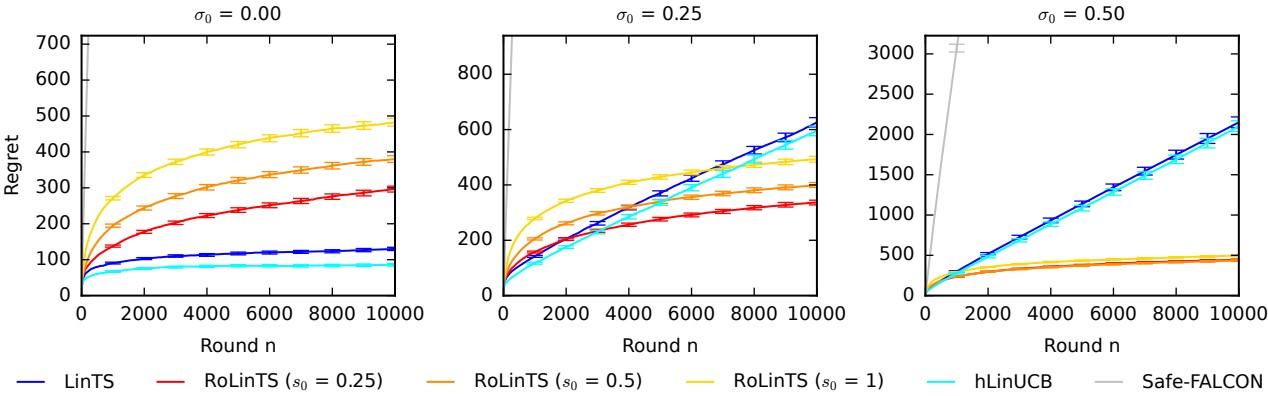

Figure 1: Comparison of `RoLinTS` to three baselines on synthetic problems in Section 7.1. The results are averaged over 100 runs.

# 7 Experiments

We conduct three main experiments. In Section 7.1, we evaluate `RoLinTS` in synthetic bandit problems. In Section 7.2, we apply it to the problem of learning a linear model with misspecified features. In Section 7.3, we compare a naive implementation of `RoLinTS` to that in Section 4.3.

We conduct four additional experiments in Appendix C. In Figure 3, we study the robustness of `RoLinTS` to parameter misspecifications. In Figure 4, we evaluate `RoLinUCB`. In Figure 5, we study how the regret of `RoLinTS` increases with the numbers of arms $K$. Finally, in Figure 6, we compare to Ghosh et al. (2017) on non-contextual problems.

## 7.1 Synthetic Experiment

Our first experiment is with three robust contextual bandits where $\sigma_0 \in \{0, 0.25, 0.5\}$. Note that $\sigma_0 = 0$ corresponds to a contextual linear bandit. We set $K = 50$ and $d = 10$. The model parameter and features are sampled from $\mathcal{N}(\mathbf{0}, \mathbf{I}_d)$ and uniformly at random from $[-1, 1]^d$, respectively. The reward noise is $\sigma = 1$. We compare `RoLinTS`, run with $\sigma_0 \in \{0.25, 0.5, 1\}$, to `LinTS`, which can be viewed as `RoLinTS` with $\sigma_0 = 0$ (Section 4.2). To distinguish $\sigma_0$ in `RoLinTS` from the environment parameter, we denote the former by $s_0$. We consider two additional baselines, `Safe-FALCON` of Krishnamurthy et al. (2021) and `hLinUCB` of Wang et al. (2016), which are introduced in Section 2. We set the complexity term in `Safe-FALCON` to $\xi(n, \zeta) = d \log(1/\zeta)/n$, since we have $d$-dimensional regression problems. In `hLinUCB`, we learn $K$ additional features per arm. This choice is motivated by the fact that `RoLinTS` can be implemented inefficiently as `LinTS` with $K$ additional features per arm (Section 6.2).

Our results are reported in Figure 1. In the left plot, the environment is a contextual linear bandit and `LinTS` has a sublinear regret. In this case, `RoLinTS` is not expected to outperform it, because it learns an additional random-effect parameter per arm. Nevertheless, all variants of `RoLinTS` have a sublinear regret in $n$. A higher regret corresponds to higher values of $s_0$, which is expected since `RoLinTS` with a higher value of $s_0$ is more uncertain about the underlying model being linear. In the middle plot, the environment is a robust contextual bandit with $\sigma_0 = 0.25$. Although this model is only slightly misspecified, `LinTS` fails and has a linear regret. In comparison, all variants of `RoLinTS` have a sublinear regret, even with the misspecified random effect $s_0 \in \{0.5, 1\}$. This highlights the robustness of our approach to misspecification. Finally, in the right plot, the environment is a robust contextual bandit with $\sigma_0 = 0.5$. We observe that the gap in the regret of `LinTS` and `RoLinTS` increases with $\sigma_0$. In this case, `RoLinTS` has at least twice lower regret than `LinTS` for all values of $s_0$. When the model is correctly specified ($\sigma_0 = 0$), `hLinUCB` outperforms both `LinTS` and `RoLinTS`. When the model is misspecified ($\sigma_0 > 0$), `hLinUCB` performs similarly to `LinTS` and `RoLinTS` outperforms it. `Safe-FALCON` is too conservative to be competitive. In all experiments, its regret at $n = 10\,000$ rounds is an order of magnitude higher than that of `RoLinTS`.

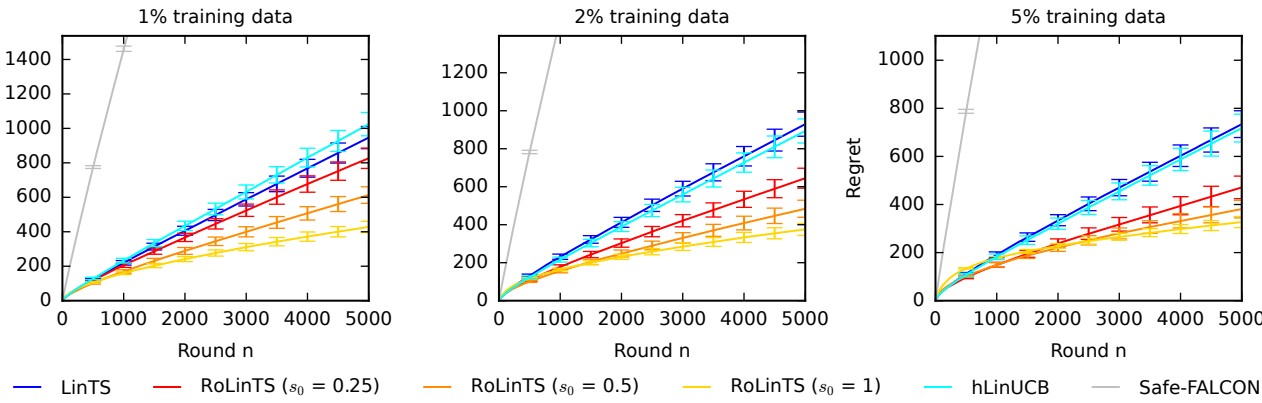

Figure 2: Comparison of `RoLinTS` to three baselines on a movie recommendation problem with misspecified item features. The results are averaged over $500$ runs.

## 7.2  MovieLens Experiment

This experiment shows the utility of `RoLinTS` in a linear bandit with misspecified features. Specifically, we have a linear function $u^\top v$ that models the mean rating of user $u$ for movie $v$, where $u$ is an unknown preference vector of the user and $v$ is a feature vector of a movie. The challenge is that the agent only knows $\hat{v}$, an estimate of $v$ from logged data. All parameters in this experiment are estimated by matrix completion: $u$ and $v$ are learned from the test set, and $\hat{v}$ is learned from the training set. When the training set is small, $\hat{v}$ is a poor estimate of $v$, and thus the mean rating of a movie is not linear in $\hat{v}$. This can be addressed by learning a separate bias term per movie, which is what `RoLinTS` does.

The experiment is specifically set up as follows. We take the MovieLens 1M dataset (Lam & Herlocker, 2016) with $n_u = 6\,000$ users, $n_i = 4\,000$ items, and $1$ million ratings. We divide the dataset equally into the training and test sets. In the training set, we apply alternating least-squares to complete the rating matrix. The result are latent user $\hat{\mathbf{U}} \in \mathbb{R}^{n_u \times d}$ and item $\hat{\mathbf{V}} \in \mathbb{R}^{n_i \times d}$ factors, where $d$ is the factorization rank and $\hat{\mathbf{U}}\hat{\mathbf{V}}^\top$ estimates the mean ratings for all user-item pairs. We apply the same approach to the test set, and obtain latent user $\mathbf{U} \in \mathbb{R}^{n_u \times d}$ and item $\mathbf{V} \in \mathbb{R}^{n_i \times d}$ factors.

The interaction with a recommender system is simulated as follows. We choose a random user and $K = 50$ random movies, and the goal is to learn to recommend the best of these movies to the chosen user. This is repeated $500$ times. The feature vector of movie $i$ is $\hat{\mathbf{V}}_{i,:}$, which is the $i$-th row of matrix $\hat{\mathbf{V}}$. The challenge is that the mean rating of movie $i$ for user $j$ is $\mathbf{U}_{j,:}\mathbf{V}_{i,:}^\top$. Therefore, it is linear in the unobserved $\mathbf{V}_{i,:}$ but not in the observed $\hat{\mathbf{V}}_{i,:}$. `RoLinTS` can adapt to this misspecification by essentially learning $\mathbf{U}_{j,:}(\mathbf{V}_{i,:} - \hat{\mathbf{V}}_{i,:})^\top$. The rating noise is $\mathcal{N}(0, \sigma^2)$ with $\sigma = 0.759$, which is estimated from data.

Our results are reported in Figure 2. In the left plot, we only use $1\%$ of the training set to estimate movie features $\hat{\mathbf{V}}$. In this case, the estimated features are highly uncertain and the regret of `LinTS` is clearly linear $n$. The regret of `RoLinTS` is significantly lower for all $s_0 \in \{0.25, 0.5, 1\}$, up to twice for $s_0 = 1$ at $n = 5\,000$. This clearly shows that `RoLinTS` can partially address the problem of misspecified features. In the next two plots, we use $2\%$ and $5\%$ of the training set to estimate movie features $\hat{\mathbf{V}}$. As the features become more precise, all methods improve. Nevertheless, the benefit of adapting to model misspecification persists. We also observe that `hLinUCB` performs better than `LinTS` but is worse than `RoLinTS`. `Safe-FALCON` is too conservative to be competitive. Its regret at $n = 5\,000$ rounds is an order of magnitude higher than that of `RoLinTS`.

## 7.3  Run Time

The challenge with implementing `RoLinTS` naively (Section 6.2) is $O((d+K)^3)$ computational cost, due to inverting $(d+K) \times (d+K)$ precision matrices. Our efficient implementation (Section 4.3) inverts only $d \times d$ precision matrices and its cost is $O(d^2(d+K))$ per round. To show this empirically, we compare the run time of `RoLinTS` to its naive

| $K$ | RoLinTS | LinTS |
|------|---------|-------|
| 16 | 0.7 | 1.1 |
| 32 | 0.8 | 2.0 |
| 64 | 0.9 | 4.6 |
| 128 | 1.1 | 14.7 |
| 256 | 1.5 | 75.6 |
| 512 | 2.9 | 498.7 |
| 1 024 | 5.2 | 4 275.4 |

Table 1: Run times of RoLinTS and LinTS as functions of the number of arms $K$. The time is measured in seconds.

implementation, called LinTS for simplicity. We take the setup from Figure 1, vary $K$ from 16 to 1024, and measure the run time of 100 random runs over a horizon of $n = 500$ rounds.

Our results are reported in Table 1. These results confirm our expectation. For large $K$, the run time of RoLinTS doubles when $K$ doubles. Therefore, it is linear in $K$. On the hand, for large $K$, the run time of LinTS is nearly cubic in $K$. For a moderately large number of arms, $K = 128$, RoLinTS is 10 times faster than LinTS. For a large number of arms, $K = 1\,024$, RoLinTS is a thousand times faster than LinTS.

## 8 Conclusions

Model misspecification in bandits, when the optimal arm under the assumed model is not optimal, can lead to catastrophic failures of contextual bandit algorithms and linear regret. We mitigate this by proposing robust contextual linear bandits. The key idea in our model is that the mean reward of an arm is a dot product of its context and a shared model parameter, which is offset by an arm-specific variable. This approach is statistically efficient because the model parameter is shared by all arms; yet quite robust to model misspecification due to learning the arm-specific variables. We propose UCB and posterior-sampling algorithms for our setting, show how to implement them efficiency, prove regret bounds that reflect the structure of our problem, and also validate the algorithms empirically.

Our work has several limitations that can be addressed by future works. For instance, although Theorem 1 reflects some structure of our problem (Section 6.1), it is not completely satisfactory. In particular, as the inter-arm heterogeneity diminishes, $\sigma_0 \to 0$, one would expected a regret bound of $O(\sqrt{dn})$ while we get $O(\sqrt{(d + K)n})$. Proving of the improved bound seems highly non-trivial due to complex correlations of all estimated model parameters in the posterior. Another limitation of our work is that we focus on linear models. We plan to extend robust contextual bandits to generalized linear models. Finally, although we have not discussed hyper-parameter tuning and learning, we do so in Appendix D.

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

# Appendix

The appendix is organized as follows. Appendix A presents the posterior of $\mu_{i,t}$. Appendix B provides a general Bayes regret analysis, which is further demonstrated in the subsequent four sections: Appendix B.1 and Appendix B.2 delve into the Bayes regret of `LinTS`, addressing scenarios involving both an infinite and finite number of contexts. Appendix B.3 extends the Bayes regret of `LinTS` to `LinUCB`. Appendix B.4 provides the upper bound on the sum of posterior variance. Appendix C presents two supplementary experiments. Finally, Appendix D explores the empirical Bayes method for hyper-parameter selection.

# A  Posterior of $\mu_{i,t}$

**Lemma 1.** *Let $z_{i,j} \sim \mathcal{N}(0, \sigma^2)$ and $\mu_i \sim \mathcal{N}(0, \sigma_0^2)$. Assuming that $\sigma^2$ and $\sigma_0^2$ are known, and $\boldsymbol{\theta} \sim \mathcal{N}(0, \lambda \mathbf{I})$, we have that $\mu_{i,t} \mid H_t \sim \mathcal{N}(\hat{\mu}_{i,t}, \tau_{i,t}^2)$.*

*Proof.* Recall the following well-known identity. Let $Y|X = x \sim \mathcal{N}(ax + b, \sigma^2)$ and $X \sim \mathcal{N}(\mu, \sigma_x^2)$. Then

$$Y \sim \mathcal{N}(a\mu + b, a^2 \sigma_x^2 + \sigma^2).$$

Obviously, if we set $X$ to $\boldsymbol{\theta} \mid H_t$ and $Y$ to $\mu_{i,t} \mid H_t$, we can apply this result to obtain the distribution of $\mu_{i,t} \mid H_t$ from the distributions of $\mu_{i,t} \mid \boldsymbol{\theta}, H_t$ and $\boldsymbol{\theta} \mid H_t$. Simple calculation shows that $\mu_{i,t} \mid \boldsymbol{\theta}, H_t$ is a Gaussian with mean in $\tilde{\mu}_{i,t} = \mathbf{x}_{i,t}\boldsymbol{\theta} + w_{i,t}(\bar{r}_{i,t} - \bar{\mathbf{x}}_{i,t}^\top \boldsymbol{\theta})$ and variance in $\sigma_0^2(1 - w_{i,t})$.

Now we derive the distribution of $\boldsymbol{\theta} \mid H_t$. Assuming $\boldsymbol{\theta} \sim \mathcal{N}(0, \lambda \mathbf{I})$, $\bar{r}_{i,t}$ can be considered to be generated from the following Bayesian model:

$$\bar{r}_{i,t} \mid \boldsymbol{\theta} \sim \mathcal{N}(\mathbf{x}_{i,t}^\top \boldsymbol{\theta}, \sigma_0^2 + \sigma^2/n_k), \quad \boldsymbol{\theta} \sim \mathcal{N}(0, \lambda \mathbf{I}).$$

Thus, the distribution of $\boldsymbol{\theta} \mid H_t$ is easily obtained as

$$\boldsymbol{\theta} \mid H_t \sim \mathcal{N}(\hat{\boldsymbol{\theta}}_t, \mathbf{Q}_t),$$

where

$$\hat{\boldsymbol{\theta}}_t = \left( \lambda \mathbf{I} + \sum_{i=1}^{K} \mathbf{X}_{i,t}^{\top} \mathbf{V}_{i,t}^{-1} \mathbf{X}_{i,t} \right)^{-1} \sum_{i=1}^{K} \mathbf{X}_{i,t}^{\top} \mathbf{V}_{i,t}^{-1} \mathbf{r}_{i,t}, \text{ and}$$

$$\mathbf{Q}_t = \left( \lambda \mathbf{I} + \sum_{i=1}^{K} \mathbf{X}_{i,t}^{\top} \mathbf{V}_{i,t}^{-1} \mathbf{X}_{i,t} \right)^{-1}.$$

Using the above results, we obtain the distribution of $\mu_{i,t} \mid H_t$ from the distributions of $\mu_{i,t} \mid \boldsymbol{\theta}, H_t$ and $\boldsymbol{\theta} \mid H_t$. That distribution is a Gaussian with mean in (14) and variance in (15). This completes the proof. □

# B    General Bayes Regret Analysis

We study a linear bandit in $d$ dimensions with action set $\mathcal{A} \subseteq \mathbb{R}^d$. The model parameter is $\theta_* \in \mathbb{R}^d$ and we assume that it is drawn from a prior as $\theta_* \sim \mathcal{N}(\theta_0, \Sigma_0)$. The mean reward of action $a \in \mathcal{A}$ is $a^\top \theta_*$. In round $t$, the learning agent takes action $A_t \in \mathcal{A}_t$, where $\mathcal{A}_t \subseteq \mathcal{A}$ is the set of feasible actions in round $t$. The round-dependent action set $\mathcal{A}_t$ can be used to model context. After the agent takes action $A_t$, it observes its noisy reward $Y_t = A_t^\top \theta_* + \varepsilon_t$, where $\varepsilon_t \sim \mathcal{N}(0, \sigma^2)$ is an independent Gaussian noise.

The history in round $t$ is $H_t$ and the posterior distribution is $\mathcal{N}(\hat{\theta}_t, \hat{\Sigma}_t)$, where

$$\hat{\theta}_t = \hat{\Sigma}_t \left( \Sigma_0^{-1} \theta_0 + \sigma^{-2} \sum_{\ell=1}^{t-1} A_\ell Y_\ell \right) , \quad \hat{\Sigma}_t^{-1} = \Sigma_0^{-1} + G_t , \quad G_t = \sigma^{-2} \sum_{\ell=1}^{t-1} A_\ell A_\ell^\top .$$

The optimal action in round $t$ is $A_{t,*} = \arg\max_{a \in \mathcal{A}_t} a^\top \theta_*$ and our performance metric is the *n-round Bayes regret*

$$R(n) = \mathbb{E}\left[ \sum_{t=1}^{n} A_{t,*}^\top \theta_* - A_t^\top \theta_* \right] ,$$

where the expectation in $R(n)$ is over random observations $Y_t$, random actions $A_t$, and random model parameter $\theta_*$.

We analyze two algorithms: linear Thompson sampling (`LinTS`) and Bayesian `LinUCB`. `LinTS` is implemented as follows. In round $t$, it samples the model parameter as $\theta_t \sim \mathcal{N}(\hat{\theta}_t, \hat{\Sigma}_t)$ and then takes action $A_t = \arg\max_{a \in \mathcal{A}_t} a^\top \theta_t$. Bayesian `LinUCB` is implemented as follows. In round $t$, it computes a UCB for each action $a \in \mathcal{A}_t$ as $U_t(a) = a^\top \hat{\theta}_t + \alpha \|a\|_{\hat{\Sigma}_t}$ and then takes action $A_t = \arg\max_{a \in \mathcal{A}_t} U_t(a)$, where $\|a\|_M = \sqrt{a^\top M a}$ and $\alpha > 0$ is a tunable parameter.

The final regret bound is stated below.

**Theorem 2.** *For any $\delta > 0$, the $n$-round Bayes regret of both* `LinTS` *and Bayesian* `LinUCB` *with* $\alpha = \sqrt{2d \log(1/\delta)}$ *is bounded as*

$$R(n) \le \sigma_{\max} d \sqrt{\frac{2n}{\log(1 + \sigma^{-2}\sigma_{\max}^2)} \log\left(1 + \frac{\sigma_{\max}^2 n}{\sigma^2 d}\right) \log(1/\delta)} + \sqrt{2/\pi}\, \sigma_{\max} d^{\frac{3}{2}} n \delta ,$$

*where $\sigma_{\max} = \max_{a \in \mathcal{A},\, t \in [n]} \|a\|_{\hat{\Sigma}_t}$. Moreover, when $|\mathcal{A}_t| \le K$ holds in all rounds $t$, the $n$-round Bayes regret of both* `LinTS` *and Bayesian* `LinUCB` *with* $\alpha = \sqrt{2 \log(1/\delta)}$ *is bounded as*

$$R(n) \le \sigma_{\max} \sqrt{\frac{2dn}{\log(1 + \sigma^{-2}\sigma_{\max}^2)} \log\left(1 + \frac{\sigma_{\max}^2 n}{\sigma^2 d}\right) \log(1/\delta)} + \sqrt{2/\pi}\, \sigma_{\max} K n \delta .$$

## B.1    Infinite Number of Contexts

We start the analysis with a useful lemma for `LinTS`.

**Lemma 2.** *For any $\delta > 0$, the $n$-round Bayes regret of* `LinTS` *is bounded as*

$$R(n) \le \sqrt{2dn\mathcal{V}(n)\log(1/\delta)} + \sqrt{2/\pi}\, \sigma_{\max} d^{\frac{3}{2}} n \delta ,$$

*where $\sigma_{\max} = \max_{a \in \mathcal{A},\, t \in [n]} \|a\|_{\hat{\Sigma}_t}$ and $\mathcal{V}(n) = \mathbb{E}\left[ \sum_{t=1}^{n} \|A_t\|_{\hat{\Sigma}_t}^2 \right]$.*

*Proof.* Fix round $t$. Since $\hat{\theta}_t$ is a deterministic function of $H_t$, and $A_{t,*}$ and $A_t$ are i.i.d. given $H_t$, we have

$$\mathbb{E}\left[ A_{t,*}^\top \theta_* - A_t^\top \theta_* \right] = \mathbb{E}\left[ \mathbb{E}\left[ A_{t,*}^\top (\theta_* - \hat{\theta}_t) \,\Big|\, H_t \right] \right] + \mathbb{E}\left[ \mathbb{E}\left[ A_t^\top (\hat{\theta}_t - \theta_*) \,\Big|\, H_t \right] \right] . \tag{20}$$

Now note that $\hat{\theta}_t - \theta_*$ is a zero-mean random vector independent of $A_t$, and thus $\mathbb{E}\left[A_t^\top(\hat{\theta}_t - \theta_*) \,\middle|\, H_t\right] = 0$. So we only need to bound the first term in (20). Let

$$E_t = \left\{ \|\theta_* - \hat{\theta}_t\|_{\hat{\Sigma}_t^{-1}} \le \sqrt{2d \log(1/\delta)} \right\}$$

be the event that a high-probability confidence interval for the model parameter $\theta_*$ in round $t$ holds. Fix history $H_t$. Then by the Cauchy-Schwarz inequality,

$$
\begin{aligned}
\mathbb{E}\left[A_{t,*}^\top(\theta_* - \hat{\theta}_t) \,\middle|\, H_t\right] &\le \mathbb{E}\left[\|A_{t,*}\|_{\hat{\Sigma}_t}\|\theta_* - \hat{\theta}_t\|_{\hat{\Sigma}_t^{-1}} \,\middle|\, H_t\right] \hfill (21)\\
&= \mathbb{E}\left[\|A_{t,*}\|_{\hat{\Sigma}_t}\|\theta_* - \hat{\theta}_t\|_{\hat{\Sigma}_t^{-1}}\mathbb{1}\{E_t\} \,\middle|\, H_t\right] + \mathbb{E}\left[\|A_{t,*}\|_{\hat{\Sigma}_t}\|\theta_* - \hat{\theta}_t\|_{\hat{\Sigma}_t^{-1}}\mathbb{1}\{\bar{E}_t\} \,\middle|\, H_t\right]\\
&\le \sqrt{2d\log(1/\delta)}\,\mathbb{E}\left[\|A_{t,*}\|_{\hat{\Sigma}_t} \,\middle|\, H_t\right] + \underbrace{\max_{a\in\mathcal{A}}\|a\|_{\hat{\Sigma}_t}}_{\le \sigma_{\max}}\mathbb{E}\left[\|\theta_* - \hat{\theta}_t\|_{\hat{\Sigma}_t^{-1}}\mathbb{1}\{\bar{E}_t\} \,\middle|\, H_t\right]\\
&= \sqrt{2d\log(1/\delta)}\,\mathbb{E}\left[\|A_t\|_{\hat{\Sigma}_t} \,\middle|\, H_t\right] + \sigma_{\max}\mathbb{E}\left[\|\theta_* - \hat{\theta}_t\|_{\hat{\Sigma}_t^{-1}}\mathbb{1}\{\bar{E}_t\} \,\middle|\, H_t\right].
\end{aligned}
$$

The second equality follows from the observation that $\hat{\Sigma}_t$ is a deterministic function of $H_t$, and that $A_{t,*}$ and $A_t$ are i.i.d. given $H_t$. Now we focus on the second term above. First, note that

$$\|\theta_* - \hat{\theta}_t\|_{\hat{\Sigma}_t^{-1}} = \|\hat{\Sigma}_t^{-\frac{1}{2}}(\theta_* - \hat{\theta}_t)\|_2 \le \sqrt{d}\|\hat{\Sigma}_t^{-\frac{1}{2}}(\theta_* - \hat{\theta}_t)\|_\infty.$$

By definition, $\theta_* - \hat{\theta}_t \mid H_t \sim \mathcal{N}(\mathbf{0}, \hat{\Sigma}_t)$, and thus $\hat{\Sigma}_t^{-\frac{1}{2}}(\theta_* - \hat{\theta}_t) \mid H_t$ is a $d$-dimensional standard normal variable. In addition, note that $\bar{E}_t$ implies $\|\hat{\Sigma}_t^{-\frac{1}{2}}(\theta_* - \hat{\theta}_t)\|_\infty \ge \sqrt{2\log(1/\delta)}$. Finally, we combine these facts with a union bound over all entries of $\hat{\Sigma}_t^{-\frac{1}{2}}(\theta_* - \hat{\theta}_t) \mid H_t$, which are standard normal variables, and get

$$\mathbb{E}\left[\|\hat{\Sigma}_t^{-\frac{1}{2}}(\theta_* - \hat{\theta}_t)\|_\infty \mathbb{1}\{\bar{E}_t\} \,\middle|\, H_t\right] \le 2\sum_{i=1}^{d}\frac{1}{\sqrt{2\pi}}\int_{u=\sqrt{2\log(1/\delta)}}^{\infty} u\exp\left[-\frac{u^2}{2}\right]\mathrm{d}u = \sqrt{\frac{2}{\pi}}d\delta.$$

Now we combine all inequalities and have

$$\mathbb{E}\left[A_{t,*}^\top(\theta_* - \hat{\theta}_t) \,\middle|\, H_t\right] \le \sqrt{2d\log(1/\delta)}\,\mathbb{E}\left[\|A_t\|_{\hat{\Sigma}_t} \,\middle|\, H_t\right] + \sqrt{\frac{2}{\pi}}\sigma_{\max}d^{\frac{3}{2}}\delta.$$

Since the above bound holds for any history $H_t$, we combine everything and get

$$
\begin{aligned}
\mathbb{E}\left[\sum_{t=1}^{n}A_{t,*}^\top\theta_* - A_t^\top\theta_*\right] &\le \sqrt{2d\log(1/\delta)}\,\mathbb{E}\left[\sum_{t=1}^{n}\|A_t\|_{\hat{\Sigma}_t}\right] + \sqrt{\frac{2}{\pi}}\sigma_{\max}d^{\frac{3}{2}}n\delta\\
&\le \sqrt{2dn\log(1/\delta)}\sqrt{\mathbb{E}\left[\sum_{t=1}^{n}\|A_t\|_{\hat{\Sigma}_t}^2\right]} + \sqrt{\frac{2}{\pi}}\sigma_{\max}d^{\frac{3}{2}}n\delta.
\end{aligned}
$$

The last step uses the Cauchy-Schwarz inequality and the concavity of the square root. This completes the proof. $\square$

## B.2 Finite Number of Contexts

The proof of Lemma 2 relies on confidence intervals that hold for any context. Further improvements, from $O(\sqrt{d})$ to $O(\sqrt{\log K})$, are possible when the number of contexts is $|\mathcal{A}_t| \le K$. This is due to improving the first term in (21). Specifically, fix round $t$ and let

$$E_{t,a} = \left\{ |a^\top(\theta_* - \hat{\theta}_t)| \le \sqrt{2\log(1/\delta)}\|a\|_{\hat{\Sigma}_t} \right\}$$

be the event that a high-probability confidence interval for action $a \in \mathcal{A}_t$ in round $t$ holds. Then we have

$$\mathbb{E}\left[A_{t,*}^\top(\theta_* - \hat{\theta}_t) \,\middle|\, H_t\right] \leq \sqrt{2\log(1/\delta)}\, \mathbb{E}\left[\|A_{t,*}\|_{\hat{\Sigma}_t} \,\middle|\, H_t\right] + \mathbb{E}\left[A_{t,*}^\top(\theta_* - \hat{\theta}_t)\mathbb{1}\left\{\bar{E}_{t,A_{t,*}}\right\} \,\middle|\, H_t\right].$$

Now note that for any action $a$, $a^\top(\theta_* - \hat{\theta}_t)/\|a\|_{\hat{\Sigma}_t}$ is a standard normal variable. It follows that

$$\mathbb{E}\left[A_{t,*}^\top(\theta_* - \hat{\theta}_t)\mathbb{1}\left\{\bar{E}_{t,A_{t,*}}\right\} \,\middle|\, H_t\right] \leq 2\sum_{a\in\mathcal{A}_t}\|a\|_{\hat{\Sigma}_t}\frac{1}{\sqrt{2\pi}}\int_{u=\sqrt{2\log(1/\delta)}}^{\infty} u\exp\left[-\frac{u^2}{2}\right]\mathrm{d}u \leq \sqrt{\frac{2}{\pi}}\sigma_{\max}K\delta\,,$$

where $\sigma_{\max}$ is defined as in (21). The rest of the proof is as in Lemma 2 and leads to the regret bound below.

**Lemma 3.** *For any $\delta > 0$, the $n$-round Bayes regret of* LinTS *is bounded as*

$$R(n) \leq \sqrt{2n\mathcal{V}(n)\log(1/\delta)} + \sqrt{2/\pi}\sigma_{\max}Kn\delta\,,$$

*where $\sigma_{\max}$ and $\mathcal{V}(n)$ are defined as in Lemma 2.*

### B.3 Bayesian LinUCB

This section generalizes Lemmas 2 and 3 to Bayesian LinUCB. The key observation is that the decomposition in (20) can be replaced with

$$\mathbb{E}\left[A_{t,*}^\top\theta_* - A_t^\top\theta_*\right] \leq \mathbb{E}\left[\mathbb{E}\left[A_{t,*}^\top\theta_* - U_t(A_{t,*}) \,\middle|\, H_t\right]\right] + \mathbb{E}\left[\mathbb{E}\left[U_t(A_t) - A_t^\top\theta_* \,\middle|\, H_t\right]\right], \tag{22}$$

where $U_t(A_t) \geq U_t(A_{t,*})$ holds by the design of Bayesian LinUCB. The second term in (22) can be rewritten as

$$\mathbb{E}\left[U_t(A_t) - A_t^\top\theta_* \,\middle|\, H_t\right] = \mathbb{E}\left[A_t^\top(\hat{\theta}_t - \theta_*) + \alpha\|A_t\|_{\hat{\Sigma}_t} \,\middle|\, H_t\right] = \alpha\mathbb{E}\left[\|A_t\|_{\hat{\Sigma}_t} \,\middle|\, H_t\right],$$

where the last inequality follows from $\mathbb{E}\left[A_t^\top(\hat{\theta}_t - \theta_*) \,\middle|\, H_t\right] = 0$, by the same argument as in Appendix B.1. To bound the first term in (22), we rewrite it as

$$\mathbb{E}\left[A_{t,*}^\top\theta_* - U_t(A_{t,*}) \,\middle|\, H_t\right] = \mathbb{E}\left[A_{t,*}^\top(\theta_* - \hat{\theta}_t) - \alpha\|A_{t,*}\|_{\hat{\Sigma}_t} \,\middle|\, H_t\right].$$

Now we have two options for deriving the upper bound. The first is

$$\begin{aligned}
A_{t,*}^\top(\theta_* - \hat{\theta}_t) - \alpha\|A_{t,*}\|_{\hat{\Sigma}_t} &\leq \|A_{t,*}\|_{\hat{\Sigma}_t}\|\theta_* - \hat{\theta}_t\|_{\hat{\Sigma}_t^{-1}} - \alpha\|A_{t,*}\|_{\hat{\Sigma}_t} \\
&\leq \|A_{t,*}\|_{\hat{\Sigma}_t}\|\theta_* - \hat{\theta}_t\|_{\hat{\Sigma}_t^{-1}}\mathbb{1}\left\{\|\theta_* - \hat{\theta}_t\|_{\hat{\Sigma}_t^{-1}} > \alpha\right\} \\
&\leq \sigma_{\max}\|\theta_* - \hat{\theta}_t\|_{\hat{\Sigma}_t^{-1}}\mathbb{1}\left\{\bar{E}_t\right\},
\end{aligned}$$

where $E_t$ is defined in Appendix B.1 and thus $\alpha = \sqrt{2d\log(1/\delta)}$. Following Appendix B.1, we get

$$\mathbb{E}\left[A_{t,*}^\top\theta_* - U_t(A_{t,*}) \,\middle|\, H_t\right] \leq \sqrt{2/\pi}\sigma_{\max}d^{\frac{3}{2}}\delta\,.$$

The second option is

$$A_{t,*}^\top(\theta_* - \hat{\theta}_t) - \alpha\|A_{t,*}\|_{\hat{\Sigma}_t} \leq A_{t,*}^\top(\theta_* - \hat{\theta}_t)\mathbb{1}\left\{A_{t,*}^\top(\theta_* - \hat{\theta}_t) > \alpha\|A_{t,*}\|_{\hat{\Sigma}_t}\right\} \leq A_{t,*}^\top(\theta_* - \hat{\theta}_t)\mathbb{1}\left\{\bar{E}_{t,A_{t,*}}\right\},$$

where $E_{t,a}$ is defined in Appendix B.2 and thus $\alpha = \sqrt{2\log(1/\delta)}$. Following Appendix B.2, we get

$$\mathbb{E}\left[A_{t,*}^\top\theta_* - U_t(A_{t,*}) \,\middle|\, H_t\right] \leq \sqrt{2/\pi}\sigma_{\max}K\delta\,.$$

It follows that the regret of Bayesian LinUCB is identical to that of LinTS.

### B.4 Upper Bound on the Sum of Posterior Variances

Now we bound the sum of posterior variances $\mathcal{V}(n)$ in Lemmas 2 and 3. Fix round $t$ and note that

$$\|A_t\|_{\hat{\Sigma}_t}^2 = \sigma^2 \frac{A_t^\top \hat{\Sigma}_t A_t}{\sigma^2} \le c_1 \log(1 + \sigma^{-2} A_t^\top \hat{\Sigma}_t A_t) = c_1 \log \det(I_d + \sigma^{-2} \hat{\Sigma}_t^{\frac{1}{2}} A_t A_t^\top \hat{\Sigma}_t^{\frac{1}{2}}) \tag{23}$$

for

$$c_1 = \frac{\sigma_{\max}^2}{\log(1 + \sigma^{-2} \sigma_{\max}^2)}.$$

This upper bound is derived as follows. For any $x \in [0, u]$,

$$x = \frac{x}{\log(1 + x)} \log(1 + x) \le \left( \max_{x \in [0,u]} \frac{x}{\log(1 + x)} \right) \log(1 + x) = \frac{u}{\log(1 + u)} \log(1 + x).$$

Then we set $x = \sigma^{-2} A_t^\top \hat{\Sigma}_t A_t$ and use the definition of $\sigma_{\max}$.

The next step is bounding the logarithmic term in (23), which can be rewritten as

$$\log \det(I_d + \sigma^{-2} \hat{\Sigma}_t^{\frac{1}{2}} A_t A_t^\top \hat{\Sigma}_t^{\frac{1}{2}}) = \log \det(\hat{\Sigma}_t^{-1} + \sigma^{-2} A_t A_t^\top) - \log \det(\hat{\Sigma}_t^{-1}).$$

Because of that, when we sum over all rounds, we get telescoping and the total contribution of all terms is at most

$$\sum_{t=1}^{n} \log \det(I_d + \sigma^{-2} \hat{\Sigma}_t^{\frac{1}{2}} A_t A_t^\top \hat{\Sigma}_t^{\frac{1}{2}}) = \log \det(\hat{\Sigma}_{n+1}^{-1}) - \log \det(\hat{\Sigma}_1^{-1}) = \log \det(\Sigma_0^{\frac{1}{2}} \hat{\Sigma}_{n+1}^{-1} \Sigma_0^{\frac{1}{2}})$$

$$\le d \log \left( \frac{1}{d} \operatorname{tr}(\Sigma_0^{\frac{1}{2}} \hat{\Sigma}_{n+1}^{-1} \Sigma_0^{\frac{1}{2}}) \right) = d \log \left( 1 + \frac{1}{\sigma^2 d} \sum_{t=1}^{n} \operatorname{tr}(\Sigma_0^{\frac{1}{2}} A_t A_t^\top \Sigma_0^{\frac{1}{2}}) \right)$$

$$= d \log \left( 1 + \frac{1}{\sigma^2 d} \sum_{t=1}^{n} A_t^\top \Sigma_0 A_t \right) \le d \log \left( 1 + \frac{\sigma_{\max}^2 n}{\sigma^2 d} \right).$$

This completes the proof.

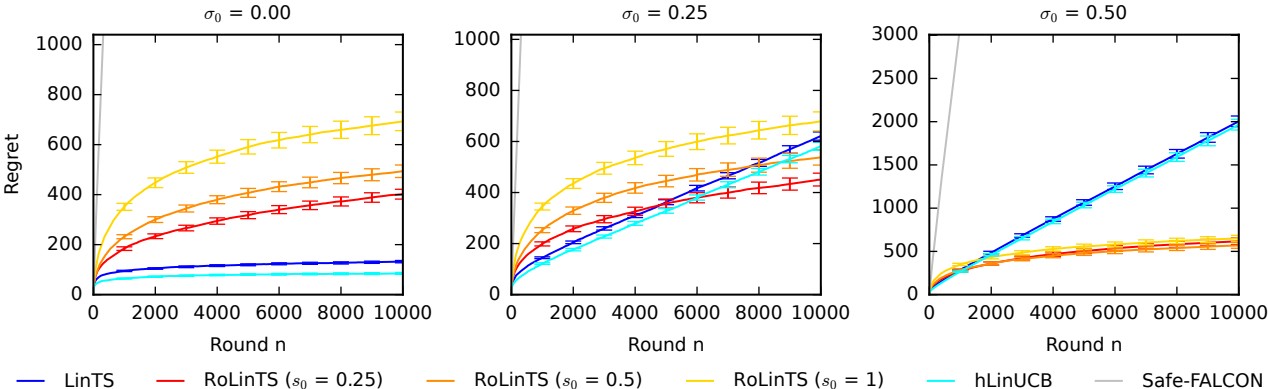

Figure 3: Comparison of misspecified `RoLinTS` to three baselines on synthetic problems in Section 7.1. The results are averaged over 100 runs.

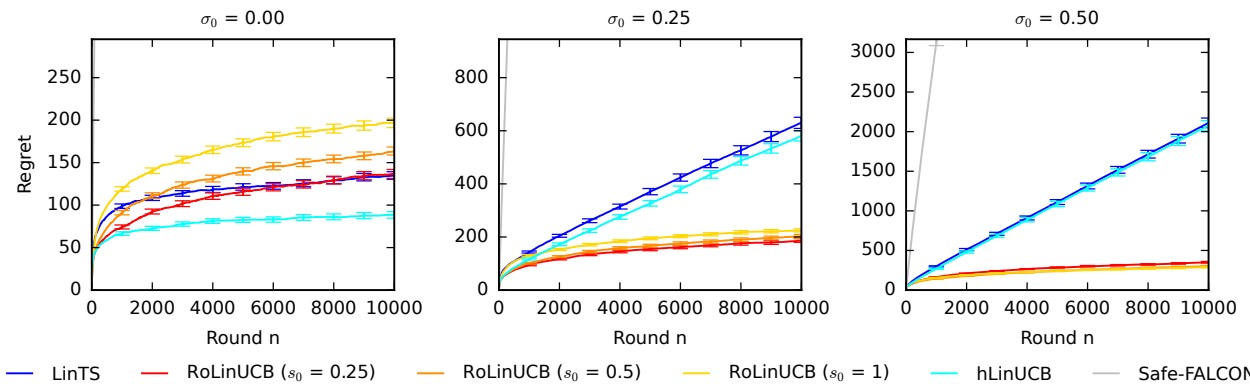

Figure 4: Comparison of `RoLinUCB` to three baselines on synthetic problems in Section 7.1. The results are averaged over 100 runs.

## C  Additional Experiments

We conduct four additional experiments on synthetic problems in Section 7.1.

In the first experiment, we randomly perturb parameters $\lambda$ and $\sigma$ of `RoLinTS` to test its robustness. For each parameter, we choose a number $u \in [1, 3]$ uniformly at random. Then we multiply it by $u$ with probability $0.5$ and divide it by $u$ otherwise. That is, the parameter is increased up to three fold or decreased up to three fold. Our results are reported in Figure 3. We observe that the regret of `RoLinTS` increases slightly. Nevertheless, all trends are similar to Figure 1. We conclude that `RoLinTS` is robust to parameter misspecification.

In the second experiment, we evaluate our UCB algorithm `RoLinUCB`. The results are reported in Figure 4. The setting is the same as in Figure 1 and we also observe similar trends. This is expected, since Bayesian UCB algorithms are known to be competitive with Thompson sampling (Kaufmann et al., 2012).

In the third experiment, we evaluate the dependence on $K$ in Theorem 1. As discussed in Section 6.1, the dependence on $\sqrt{K}$ does not vanish for any $\sigma_0 > 0$ due to the suboptimality of our proof technique. To show this, we demonstrate that `RoLinTS` with $\sigma_0 \to 0$ has a comparable regret to `LinUCB` in a linear bandit. Note that the state-of-the-art regret bounds of `LinUCB` are at most $O(\log K)$ (Lattimore & Szepesvari, 2019). The linear bandit is the synthetic problem in Section 7.1 with $\sigma_0 = 0$ and $K \in \{20, 50, 100\}$. `RoLinTS` is run with $\sigma_0 \in \{0.1, 0.05, 0.01\}$. To distinguish $\sigma_0$ in `RoLinTS` from that in the environment, we denote the former by $s_0$. Our results are reported in Figure 5. We observe that the regret of `RoLinTS` approaches that of `LinTS` as $\sigma_0 \to 0$ for all $K$. When $\sigma_0 = 0.01$, the regret of `RoLinTS` is

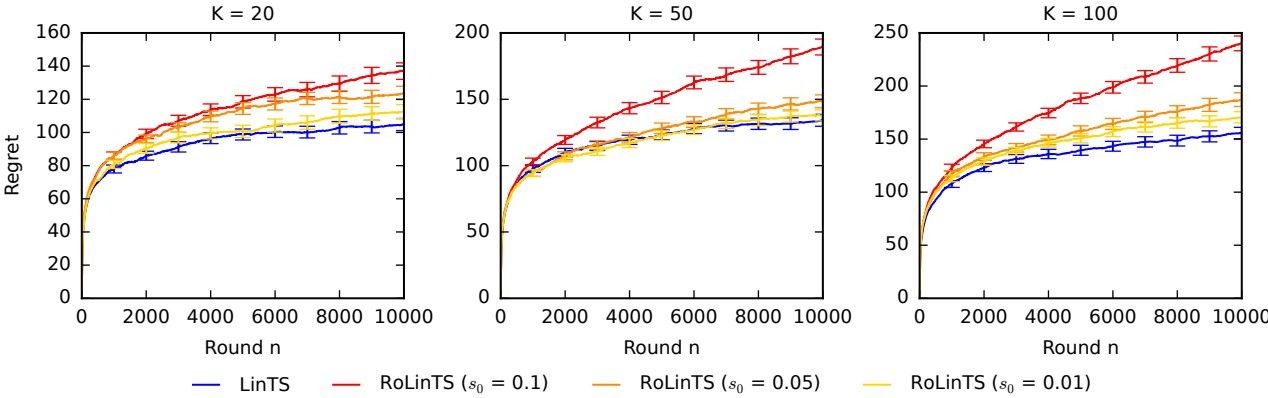

Figure 5: Regret of `RoLinTS` as a function of the numbers of arms $K$.

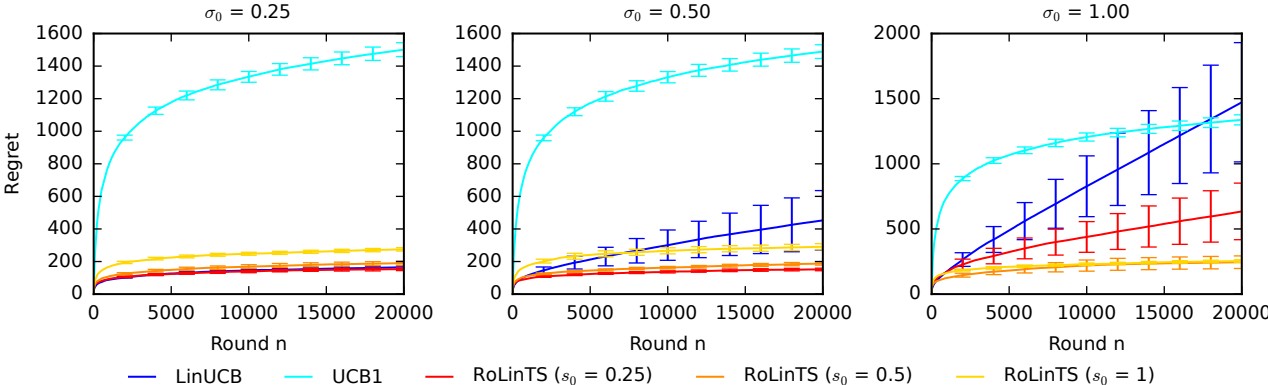

Figure 6: Comparison of `RoLinUCB` to two baselines on synthetic problems in Section 7.1. The features are sampled only once in round 1 and then kept fixed. The results are averaged over 100 runs.

almost indistinguishable from that of `LinTS`. This validates our hypothesis that the regret of `RoLinTS` does not seem to increase significantly with $K$ as $\sigma_0 \to 0$.

In the last experiment, we try to compare to Ghosh et al. (2017). Their algorithm performs an initial exploration, and then runs either `LinUCB` (Abbasi-Yadkori et al., 2011) or `UCB1` (Auer et al., 2002). A comparison to this algorithm is challenging for two reasons. First, the algorithm is non-contextual and thus would have linear regret in Figure 1. To have a more fair comparison, we experiment with non-contextual problems here. Specifically, the features of the arms in Section 7.1 are sampled only once in round $t = 1$ and then kept fixed. Second, the algorithm of Ghosh et al. (2017) has tunable parameters. Rather than fixing them, which could put the algorithm at a disadvantage, we compare to its two components: `LinUCB` and `UCB1`. If `RoLinTS` can outperform both, the algorithm of Ghosh et al. (2017) could not be competitive with `RoLinTS`. Our results are reported in Figure 6. In all problems, `UCB1` has a high regret because the number of arms is large, $K = 50$. Moreover, when the model misspecification is high, $\sigma_0 \geq 0.5$, `LinUCB` has a linear regret. Therefore, when $\sigma_0 \geq 0.5$, `RoLinTS` would outperform the algorithm of Ghosh et al. (2017).

# D    Choosing Hyper-Parameters

In this paper, we assume that the hyper-parameters are known by the agent. When the hyper-parameters are unknown, they have to be estimated. We end this paper by discussing this challenge.

For a fully-Bayesian treatment of hyper-parameters, it is desirable to marginalize over them. However, this is computationally intensive, often involving an integration. Let $\psi = (\sigma^2, \sigma_0^2, \lambda)^\top$ and $\hat{\mu}_{i,t}(\psi)$ be the prediction as a function of $\psi$. By assuming some distribution on $\psi$, the fully-Bayesian method tries to take the following integration:

$$\hat{\mu}_{i,t}^{\text{FB}} = \int \hat{\mu}_{i,t}(\psi) p(\psi|H_t) d\psi.$$

Although the integral can be done using many powerful tools in Bayesian statistics, the fully-Bayesian method is sensible to the prior setting of $\psi$, more seriously, is computationally intensive in bandit algorithms that requires sequential updating.

Compared to a fully-Bayesian treatment, a simpler method is the empirical Bayes method (Carlin & Louis, 2000), which plugs in the estimates of the hyper-parameters from data. Various methods of obtaining consistent estimators are available, including the method of moments, maximum likelihood, and restricted maximum likelihood (Harville, 1988). Here we adopt the method of moments, since it has explicit formulas to update each round.

Unbiased quadratic estimate of $\sigma^2$ is given by

$$\hat{\sigma}^2 = \left( \sum_{i=1}^{K} (n_{i,t} - d) \right)^{-1} \sum_{i=1}^{K} \sum_{\ell \in \mathcal{T}_{i,t}} \hat{e}_{i,\ell}^2,$$

where $n_t = \sum_{i=1}^{K} n_{i,t}$, and $\hat{e}_{i,\ell}$ are the residuals for the regression of the $r$ deviation, $r_{i,\ell} - \bar{r}_{i,t}$, on the context $\mathbf{x}$ deviations, $\mathbf{x}_{i,\ell} - \bar{\mathbf{x}}_{i,t}$, for those rewards with $n_{i,t} > 1$. Unbiased quadratic estimate of $\sigma_0^2$ is given by

$$\tilde{\sigma}_0^2 = (n_t - d_0)^{-1} \left[ \sum_{i=1}^{K} \sum_{\ell \in \mathcal{T}_{i,t}} \hat{s}_{i,\ell}^2 - (n_t - d)\hat{\sigma}^2 \right],$$

where $d_0 = (\sum_{i=1}^{K} \mathbf{X}_{i,t}^\top \mathbf{X}_{i,t})^{-1} \sum_{i=1}^{K} n_{i,t}^2 \bar{\mathbf{x}}_{i,t} \bar{\mathbf{x}}_{i,t}^\top$ and $\hat{s}_{i,\ell}$ are the residuals for the regression of $r_{i,\ell}$ on the context $\mathbf{x}_{i,\ell}$. It is possible that $\tilde{\sigma}_0^2$ is negative. Thus, we define $\hat{\sigma}_0^2 = \max\{\tilde{\sigma}_0^2, 0\}$. For simplifying the choice of $\lambda$, it is not bad to let $\lambda$ equal a tiny constant, such as $\lambda = 0.01$.

Although the empirical Bayes method with plugged-in variance estimates is convenient and useful, it is challenging to analyze. The reason is that the randomness in the estimated hyper-parameters needs to be considered in the regret analysis. We leave the theoretical investigation of this challenge as an open question of interest.

