# OpenReview forum: "Robust Contextual Linear Bandits"
_TMLR — Rejected by TMLR_

### Review · Reviewer_3vtJ · 2023-08-06

**Summary Of Contributions:**

This paper considers the linear contextual bandit problem, in a mis-specified model setting. In particular, the authors focus on a setting where the linear combination of arm-dependent context and some common regression parameters does not (alone) explain the expected reward of an arm, but that there are some additional arm-dependent terms which account for variability in the rewards. The central challenge considered in the paper, is the design of algorithms which perform well both in the presence and absence of such additional arm-dependent terms and are therefore robust to model mis-specification regarding these parameters.

The paper proposes UCB and Thompson Sampling (TS) policies for the problem, provides a regret analysis for both approaches, demonstrates that the computational complexity is reduced compared with reasonable competitors, and demonstrates the competitive empirical performance of the proposed algorithms (particularly TS).


**Audience:**

Yes

**Broader Impact Concerns:**

I have no concerns I feel ought to be addressed.

**Claims And Evidence:**

Yes

**Requested Changes:**

Critical for Acceptance:

-	I feel it is remiss not to offer an empirical comparison to the work of Ghosh et al. (2017). While the submission does note the paper and describe how the algorithm is differently structured, it is nevertheless applicable to the problem and it is of interest to the reader how much worse (or for which horizons) this method is.
-	I feel that to include the proposing of the model as a contribution of the paper is an overstatement. As far as I can tell it is the same model as in Ghosh et al. and other earlier works, just with an explicit focus on the regret minimisation objective. If I have understood correctly, I think this point should not be obfusciated by claiming the model as a contribution.
-	Most notably, the submission does not mention Takemura et al. (2021, AISTATS) which as far as I can tell, considers the same problem only under an assumption of bounded noise, rather than the (slightly) more general sub-Gaussian case here. It does use a fairly different algorithm, built on SupLinUCB rather than LinUCB. The major limitation of the manuscript is its lack of acknowledgement of this work, and I would ask that the authors carefully describe in their reply/revision how the submission advances the literature beyond Takemura et al.’s work, with the view that this is the most critical change. As a sub-point to this, the works of Dong and Yang (2023, ICML) and Zhang et al (2023, arxiv) which I accept are more realistically contemporaneous would be nice to work in to the discussion of related literature, as they both seem to provide useful contributions to the wider theoretical context of the problem.

Simply Strengthening the Work:

-	The explanation in the paragraph at the end of page 1 would be stronger if it gave an example of what an arm is interpreted to be in this setting.
-	Three lines before end of section 1: I think you mean ‘statistically’ not ‘statically’?
-	Line 1 of page 3: I would consider being specific here that you are considering linear contextual bandits, and perhaps even further that you work with (sub-)Gaussian noise, rather than present in generality here but clarify that all results are for a specific case later.
-	I think the Appendix would benefit from some initial discussion of its structure for completeness – which sections contribute the proofs of Theorem 2 etc, and what the various subsections cover.
-	I don’t think the statement of Theorem 2 needs the comment about $\mathcal{V}(n)$, perhaps that was copied over from Lemma 2.

References

-	Ghosh, Chowdhury, and Gopalan (2017) ‘Misspecified Linear Bandits’ AAAI.
-	Takemura, Ito, Hatano et al. (2021) ‘A Parameter Free Algorithm for Misspecified Linear Contextual Bandits’ AISTATS.
-	Dong and Yang (2023) ‘Does Sparsity Help in Learning Misspecified Linear Bandits?’ ICML
-	Zhang, He, Fan, and Gu (2023) ‘On the Interplay Between Misspecification and Sub-Optimality Gap in Linear Contextual Bandits’ arxiv:2303.09390


**Strengths And Weaknesses:**

Summary: The main strengths of the paper are in its clarity (of writing and objective) and accuracy (of the technical work). Its main limitations are in its relationship to the existing literature (both in terms of how it cites related work, and the overall novelty of the contribution).

Strengths: The paper is easy to follow. The problem is defined clearly and the motivation in terms of practical applications and the gap in the literature is made clear. The proposed algorithms are sensible, simple and intuitive and the theoretical analysis of their regret is, as far as I can tell, without error. A sensible experimental section then follows, which demonstrates the efficacy of the proposed TS approach (which is to be expected to be superior to the UCB) and provides details and justification of hyperparameter choices in the appendix. All in all, placing significance aside, the core research is sound.

Weaknesses (more detail follows in requested changes): the main (fixable) weakness of the paper is in how the contributions are related to the existing literature. The methods of Ghosh et al. and Takemura et al. seem to be (more or less) applicable to this problem and are dismissed or not cited. Other than this, novelty is in itself a slight weakness of the paper. The main novelty of the method is in its reduced computational complexity, otherwise the performance is comparable to existing approaches and the theoretical analysis is, as stated, a tweak on ‘trivial linear bandit analysis’. My requested changes focus on the related literature rather than novelty, as I feel it is here that meaningful improvements can realistically be made, but if the authors feel I have missed something in my assessment of the novelty of the work, I would encourage them to say so.

---

> ### Author Response · Authors · 2023-09-13
>
> We would like to thank the reviewer for the valuable feedback. We answer your questions below. If you have any additional concerns, please reach out to us to discuss them.
>
> **Q1: Empirical comparison to Ghosh et al. (2017)**
>
> Ghosh et al. (2017) do not handle changing context and therefore their algorithm would have linear regret in all experiments in Section 7. This is why we did not compare to them initially and this was stated in Section 2. To address your comment, we added an empirical comparison in the non-contextual setting in Appendix C (Figure 6).
>
> **Q2: Our modeling contribution**
>
> We propose a two-level Bayesian model, which additionally incorporates the Bayesian view to account for inter-arm heterogeneity. We clarified this in Section 3.1. Using this approach, we can learn stochastic deviations from the linear model. In contrast, Ghosh et al (2017) choose between LinUCB and UCB1 after an initial exploration period. Our approach is clearly beneficial, as can be seen in Appendix C (Figure 6).
>
> **Q3: Related work**
>
> We expanded the discussion in Section 2 to include your suggested works.
>
> **Q4: Additional minor comments**
>
> We have modified the paper accordingly.

---

### Review · Reviewer_erFj · 2023-08-17

**Summary Of Contributions:**

The authors consider a  contextual linear bandit problem under misspecification. More precisely, the authors consider model misspecification where the reward of pulling arm $i$, consists in expectation of a linear term $x_{i}^\top \theta$ plus an additive $\nu_i$ due to model inter-heterogeneity between arms that cannot be captured by the linear term. The goal is to design an algorithm that minimizes regret while being robust to the misspecification of the linear model. The authors are specifically interested in the Bayesian regret and therefore assume that $\theta$ and $\nu_i$ are sampled from Gaussian distributions. They propose two Bayesian algorithms RoLinUCB and RoLinTS, and show that both algorithms achieve a Bayesian regret guarantee of order $\tilde{O}(\sqrt{(\sigma_0^2 + L^2/\lambda^{-1}) (d+K)n})$. The authors further provide numerical experiments to complement their theoretical findings.

**Audience:**

No

**Broader Impact Concerns:**

I don't think there are particularly societal or ethical concerns about this work to be flagged.

**Claims And Evidence:**

Yes

**Requested Changes:**

Please try to address Weaknesses **W1** and **W2**.  I believe that an instance dependent guarantee or a more refined analysis will clarify the gain from the structure considered in this problem which at the moment is not clear from the theoretical guarantees.  There might definitely be some merit in the algorithm design but such merit is yet to be established!

**Strengths And Weaknesses:**


**Strengths.** Below I describe what I think are strengths of this work
-  **S1.** Contextual linear bandit problem under misspecification is an interesting problem.
-  **S2.** Overall, the paper is easy to follow. The problem description, algorithm design, theorems and proofs are well presented.

**Weaknesses.** Below are my concerns regarding this submission
-  **W1. Regarding the Bayesian regret.** The authors focus on Bayesian regret while existing work in the bandit literature have also studied the Frequentist regret. I believe some discussion on this is needed to clarify how this work can be positioned in the literature. In other words why do the authors focus on a Bayesian regret and not a Frequentist regret? What are the advantages or disadvantages of each framework?
-  **W2. Regarding the gain from structure.** The discussion on the regret is much appreciated. However, it seems to me that the guarantees are quite weak in the sense that if one uses an algorithm meant fo an unstructured bandit setting (e.g., UCB) they will obtain a minimax regret guarantee of order $\sqrt{KT}$ (see, e.g. [1]) while the authors only get a regret of order $\sqrt{(d+K)T}$  which suggests that there is no particular gain from the structure of the problem (since $d \ll K$). The authors argue in section 6.1 that there is an improvement in the constants but in Frequentist regret, the algorithm will not depend on any of of the parameters $L$ or $\lambda$.  Some clarification is needed here! To be honest, I think an instance dependent guarantee is more suited for this type of setups. Can the author clarify further this part? There is definitely merit in the algorithm design proposed by the authors, but the guarantees do not show it.
- **W3. Regarding running time of the algorithm.** I am not sure the discussion on the running time is very relevant and deserves to be considered a contribution, though it is appreciated.


[1] T. Lattimore and C. Szepesvari, "Bandit Algorithms",  2020

---

> ### Author Response · Authors · 2023-09-13
>
> We would like to thank the reviewer for the valuable feedback. We answer your questions below. If you have any additional concerns, please reach out to us to discuss them.
>
> **W1: Bayes regret**
>
> We conduct a Bayesian analysis because complex structures among problem parameters, such as hierarchy or random effects, cannot be captured by frequentist analyses. Several papers that conduct similar analyses are discussed in the last paragraph of Section 2. The shortcoming of Bayesian analyses is that they assume a correctly specified reward model, such as Gaussian rewards. To address this limitation, we show robustness to model misspecification empirically in Appendix C.
>
> **W2: Lower regret due to structure**
>
> We added an experiment to Appendix C (Figure 5) where we show that the empirical regret of RoLinTS approaches that of LinTS as $\sigma_0 \to 0$. Therefore, the dependence on $K$ in Theorem 1 as $\sigma_0 \to 0$, through $\sqrt{d + K}$, is an artifact of the current analysis. This is because we analyze RoLinTS as LinTS with $d + K$ features.

---

### Review · Reviewer_zij3 · 2023-08-31

**Summary Of Contributions:**

This paper studies the problem of model misspecification in linear contextual bandits. The mean reward of each arm is assumed to be equal to the dot product between the context and an unknown global vector + an arm-specific unknown offset. The authors present two algorithms, RoLinUCB and RoLinTS, which are extensions of LinUCB and LinTS to the robust linear bandit. The authors present bounds that illustrate how the Bayesian regret scales w.r.t. uncertainty in the global vector and the level of inter-arm heterogeneity. Finally, they perform an empirical evaluation of the algorithms showing an improvement over baselines.

**Audience:**

Yes

**Broader Impact Concerns:**

No concerns

**Claims And Evidence:**

Yes

**Requested Changes:**

1. (Critical) I would like to see a more thorough comparison to the hybrid model in Li et al 2010. My current understanding is that your model is a special case of their hybrid model. They consider a hybrid model of the form $z_{t, a}^\top \beta + x_{t, a}^\top \theta_a$ where $\theta_a$ is an arm-specific unknown vector (See Eq 6 in Li et al). To get an arm-specific unknown parameter we set $x_{t, a}=1$ for all arms and let $\theta_a$ be an unknown scalar for each arm. This means that we let each arm have one arm-specific feature that is constant and all other features are shared by all arms.  Is there a difference between this approach and yours?

2. Some comments comparing the running time to the one stated in Li et al 2010 would also be nice to see.

3. I like that you test the robustness of RoLinTS w.r.t. misspecified parameters (Appendix A.7.) and I wonder to what extent the regret bounds hold under a misspecified prior. Are there cases where you can still guarantee sub-linear regret even though $\lambda$ and $\sigma$ are misspecified?

4. The fact that the dependence in K doesn’t vanish as $\sigma_0$ goes to $0$ is a bit unsatisfactory. Is it an artefact of the proof or a feature of the algorithms? This can, at least partially, be addressed empirically by plotting the regret of RoLinTS against $\sigma_0$ and compare to the regret of standard LinTS.

5. The legend in Fig 1 is a bit weird and continues over several plots. This makes it somewhat hard to parse the results. A suggestion is to put a global legend on top of all the subfigures to make Fig 1 cleaner.

**Strengths And Weaknesses:**

# Strengths

1.	The paper is clearly written.

2.	The setting is well-motivated.

3.	The regret bounds are clearly stated and illustrate that the proposed algorithms have an advantage over standard algorithms for linear bandits.

4.	The empirical results suggest an improvement over previous algorithms.

# Weaknesses

1.	The empirical results suggest that RoLinTS is robust to misspecification in the inter-arm heterogeneity parameter $\sigma_0$. This is nice but I’m also a bit puzzled by the result since, to me, it suggests that knowing $\sigma_0$ isn’t that important for the result and the improvement in regret comes mainly from having a realisable linear model. That is, the problem is linear with d+K parameters but LinTS is only given d parameters which explains its linear regret. Models augmenting the linear model with arm-specific vectors exists in the literature, see Li et al 2010. The authors do have a brief comment about the relation to Li et al 2010 in the paper, but I still fail to see the differences between this model and the hybrid model in Li et al 2010 (see requested changes).

2.	The dependence on K doesn’t vanish as the inter-arm heterogeneity diminish. Is this an artefact of the proof or due to something in the algorithms/estimators?

3.	The regret bounds rely on the fact that the algorithms know the true prior distribution. This is a strong assumption that might be hard to satisfy in practice (see requested changes).


Lihong Li, Wei Chu, John Langford, and Robert E. Schapire. 2010. A contextual-bandit approach to personalized news article recommendation. In Proceedings of the 19th international conference on World wide web (WWW '10). Association for Computing Machinery, New York, NY, USA, 661–670. https://doi.org/10.1145/1772690.1772758

---

> ### Author Response · Authors · 2023-09-13
>
> We would like to thank the reviewer for the valuable feedback. We answer your questions below. If you have any additional concerns, please reach out to us to discuss them.
>
> **Q1: Comparison to the hybrid model of Li et al. (2010)**
>
> The main difference from the hybrid linear model of Li et al. (2010) is that our approach is a two-level Bayesian model, which additionally incorporates the Bayesian view to account for inter-arm heterogeneity. We clarified this in Section 3.1.
>
> **Q2: Runtime comparison to Li et al. (2010)**
>
> The purpose of our runtime comparison in Section 7.3 is to show that the implementation of our method does not increase the computational cost compared to a linear contextual bandit with the same covariates. There is no significant difference in runtime when compared to Li et al. (2010) as their model also uses the same covariates.
>
> **Q3: Guarantees for model misspecification**
>
> Assuming a correctly specified model is a standard assumption in Bayesian analyses, as there are no practical guarantees for misspecification. This is a shortcoming of Bayesian analyses. To address it, we demonstrate robustness to misspecification empirically in Appendix C.
>
> **Q4: Dependence on $K$ vanishes empirically as $\sigma_0 \to 0$**
>
> We added an experiment to Appendix C (Figure 5) where we show that the empirical regret of RoLinTS approaches that of LinTS as $\sigma_0 \to 0$. Therefore, the dependence on $K$ in Theorem 1 as $\sigma_0 \to 0$ is an artifact of the current analysis.
>
> **Q5: Weird legends**
>
> This was corrected in all plots.

---

### Decision · Action_Editors · 2023-10-07

**Recommendation:** Reject

**Comment:**

The reviewers were split in their recommendations but pointed out several remaining shortcomings of the manuscript, primarily with respect to the positioning of the manuscript in relation to prior works. These issues were raised again, and emphasized, in the discussion following the authors' response and revision. We invite a submission of a revised manuscript that improves these areas of the work.

**Proposed changes:**

* The paper should better position its contributions (data model, theory, and algorithm) in relation to the existing literature on linear bandits with model misspecification and algorithms that are robust to it. Similar settings have been studied under different names in the existing literature (e.g. Ghosh et al. 2017; Takemura et al. 2021). Moreover, a regret bound of $\tilde{O}(\sqrt{(d+K)T})$, matching the proposed work up to logarithmic factors, could be achieved by running e.g., SupLinUCB on the extended d+K-dimensional context where each arm is given a time-independent context and matching parameter. The main contribution is therefore the Bayesian analysis, but this is given only minor attention in the presentation (not in the title, barely in the abstract).

* The extent to which model misspecification is emphasized in the presentation is not matched by analysis (none) or experiments (only in the appendix, and not w.r.t. reward structure). The presentation should either be changed to match the existing results or add results to match the presentation.

* It is not clear how details of the result relate to frequentist bounds; the latter are mentioned only in the text of the related work section. The proposed Bayesian bound provides insights into how the prior interacts with the regret but gives less insight into what constitutes a hard/easy bandit instance in this model, which a frequentist bound would do. The paper would benefit greatly from a more explicit comparison to previous bounds and assumptions in order to make the differences clear and also to highlight what is gained by the Bayesian analysis. Especially since there are several papers that study the same/or very similar settings, see e.g., Table 1 in Takemura et al (2021).

**Audience:**

Yes.

**Claims And Evidence:**

The positioning of this paper is that it introduces a new bandit setting called "Robust contextual linear bandits" which addresses the common problem of "model misspecification". The setting itself is claimed to be a contribution. Several reviewers opposed this view, stating similarities to multiple known works which were not properly addressed in the paper.

The proposed algorithms are claimed to be "robust to model misspecification", but the algorithms are given the extract structure of the reward model. Only a small empirical result is given in the appendix for when the heterogeneity parameter is varied. It is unclear how the  algorithms would perform under any misspecification in the reward structure. This is in contrast to previous works that focus on how estimators behave under misspecification (e.g. Ghosh et al. 2017; Takemura et al. 2021; Krishnamurthy et al ..2021).

**Resubmission Of Major Revision:**

The authors may consider submitting a major revision at a later time.